

# Quantitative precipitation estimation with weather radar using a data- and information-based approach

Malte Neuper[1], Uwe Ehret[1]

[1]Institute of Water Resources and River Basin Management, Karlsruhe Institute of Technology - KIT, Karlsruhe, Germany

*Correspondence to*: Uwe Ehret (uwe.ehret@kit.edu)

**Abstract.** In this study we propose and demonstrate a data-driven approach in an information-theoretic framework to estimate precipitation quantitatively. In this context predictive relations are expressed by empirical discrete probability distributions directly derived from data instead of fitting and applying deterministic functions as it is standard operational practice. Applying a probabilistic relation has the benefit to provide joint statements about rain rate and the related

estimation uncertainty. The information-theoretic framework furthermore allows integration of any kind of data considered useful and explicitly considers the uncertain nature of quantitative precipitation estimation (QPE). With this framework we investigate the information gains and losses associated with various data and practices typically applied in QPE. To this end we conduct six experiments using four years of data from six laser optical disdrometers, two micro rain radars MRR regular rain gauges, weather radar reflectivity and other operationally available meteorological data from existing stations. Each

experiment addresses a typical question related to QPE: First, we measure the information about ground rainfall contained in various operationally available predictors. Here weather radar proves to be the single most important source of information, which can be further improved when distinguishing radar reflectivity – ground rainfall relationships (Z-R relations) by season and prevailing synoptic circulation pattern. Second, we investigate the effect of data sample size on QPE uncertainty using different data based predictive models. It shows that the combination of reflectivity and month of the year as a double

predictor model is the best trade-off between robustness of the model and information gain. Third, we investigate the information content in spatial position by learning and applying site-specific Z-R relations. The related information gains are only moderate and especially lower than when distinguishing Z-R relations according to time of the year or synoptic circulation pattern. Fourth, we measure the information loss when fitting and using a deterministic Z-R relation, as it is standard practice in operational radar based QPE applying e.g. the standard Marshal-Palmer relation, instead of using the

empirical relation derived directly from the data. It shows that while the deterministic function captures the overall shape of the empirical relation quite well, it introduces additional 60% of uncertainty when estimating rain rate. Fifth, we investigate how much information is gained along the radar observation path, starting with reflectivity measured by radar at height, continuing with the reflectivity measured by a MRR along a vertical profile in the atmosphere and ending with the reflectivity observed by a disdrometer directly at the ground. The results reveal that considerable additional information is

gained by using observations from lower elevations by avoiding information losses caused by ongoing microphysical precipitation processes from cloud height to ground. This underlines both the importance of vertical corrections for accurate



QPE and of the required MRR observations. In the sixth experiment we evaluate the information content of radar data only, rain gauge data only and a combination of both as a function of the distance between the target and predictor rain gauge. The results show that station-only QPE outperforms radar-only QPE up to a distance of 7 to 8 km and that radar-and-gauge QPE performs best, even compared to radar-based models applying season or circulation pattern.

## 1 Introduction

Quantitative precipitation estimation in high temporal and spatial resolution and high quality are important prerequisites for many hydro-meteorological design and management purposes. Besides rain gauges with its own limitations (Huff, 1970, Nešpor and Sevruk 1999, Nystuen 1999, Yang et al. 1999), weather radar plays an increasingly important role for QPE: Radar data have among other been used for urban hydrology (Thorndal et al. 2017, Cecinate et al. 2017, Wang et al. 2015), hydrological analysis and modelling (Bronstert et al. 2017, Rossa et al. 2005), real-time QPE (Germann et al. 2006), rainfall climatology (Overeem et al. 2009), rainfall pattern analysis (Kronenberg et al. 2012, Ruiz-Villanueva et al. 2012) and rainfall frequency analysis (Goudenhoofdt et al. 2017). For a comprehensive overview on radar theory and applications, see Battan (1959a,b), Sauvageot (1992), Doviak and Zrnic, (1993), Rinehart (1991), Fabry (2015) or Rauber and Nesbitt (2018). While the advantage of weather radar is that it provides 3-dimensional observations in high spatial and temporal resolution and large coverage, unfortunately its use relies on some sometimes more, sometimes less justified assumptions and is hampered by considerable error and uncertainty arising from measuring the radar reflectivity factor Z (herein after referred to as reflectivity) instead of rain rate R, measuring at height instead of at the ground, and many other factors such as ground clutter, beam blockage, attenuation, second-trip echoes, anomalous beam propagation and bright band effects . For a good overview on sources of errors see Zawadzki,(1984) or Villarini and Krajewski (2010). In this paper we will focus on the first two aspects, namely Z-R relation and vertical profile of reflectivity. Typically, the Z-R relation is expressed by a deterministic exponential function of the form $Z = a \cdot R^b$ fitted to simultaneous observations of Z at height and R at the ground, the most common being the Marshall-Palmer relation (Marshall and Palmer 1948). Much work has since then been done to acknowledge the strongly non-linear and time-variant nature of this relation (Lee and Zawadzki 2005, Cao et al. 2010, Adirosi et al. 2016) and its transition from at height to the ground (Vignal et al. 1999, Vignal et al. 2000). Much effort has also been spent on ways to improve QPE by combining radar with other sources of information such as rain gauges (Goudenhoofdt and Delobbe 2009, Wang et al. 2015) or numerical weather prediction (Bauer et al. 2015) and to quantify uncertainty of radar-based QPE (Cecinati et al. 2017).

In this context, it is the aim of this paper to suggest and apply a framework which comprises expressing relations among data directly by empirical discrete probability distributions (dpd's), and measuring the strength of relations and remaining uncertainties with measures from Information Theory. Comparable approaches have been suggested by Sharma and Mehrotra (2014) and Thiesen et al. (2018).



In particular, we investigate the effect of applying a purely data-based, probabilistic Z-R relation instead of a deterministic function fitted to the data (see Yang et al. 2017 or Kirstetter et al. 2015 for a similar approach). The potential advantage of applying a probabilistic relation is that it yields joint statements of both the value of R and the related estimation uncertainty. We further test the potential of various operationally available observables such as synoptic Circulation Patterns (CP),

convective and other meteorological indices (retrieved from rawinsonde data), ground meteorological variables and season indicators to distinguish typical Z-R relationships. The idea is to improve QPE by applying Z-R relationships tailored to the prevailing hydro-meteorological situation as expressed by the predictors. Lastly, we use a comprehensive data set from four years of 1-h data available from one C-band weather radar, two vertical micro rain radars (MRR), six laser beam disdrometers and six rain gauges set up in the 288 km$^2$ catchment of the Attert, Luxembourg (Fig. 2) to evaluate the

information gains about ground rainfall when moving from measuring at height to measuring closer to the ground.

The remainder of the paper is structured as follows: In the next section, we briefly present the experiments carried out in the paper. In section 2, we give a short overview on concepts and measures from Information Theory (section 2.1) and on the methods (section 2.2) and data (section 2.3) used in the study. In section 3, we present and discuss the results of all experiments; our conclusions are presented in section 4.

**1.1 Design of experiments**

We conduct altogether six experiments: In *Experiment 1*, we investigate the information about ground rainfall contained in various predictors, such as weather radar observations alone or in combination with additional, operationally available hydro-meteorological predictors. In *Experiment 2*, we investigate the effect of limited data on the uncertainty of ground rainfall estimation for various data-based models. In *Experiment 3*, we examine to which degree the empirical Z-R

relationship in the 288 km² test domain varies in space, and what minimum data sets are required to support the use of site-specific Z-R relations. In *Experiment 4*, we evaluate the effect of functional compression by measuring the information loss when using the deterministic Marshall-Palmer relationship instead of the empirical, 'scattered' relationship between Z and R as contained in the data. In *Experiment 5*, we investigate information gains along the radar path, i.e. when we use observations of the reflectivity measured increasingly close to the ground. We start with observations from weather radar at

height, continue with observations from MRR along a vertical profile and finally use disdrometer observations of the reflectivity at the ground. In the last *Experiment 6*, we compare two methods of QPE, radar-based and rain gauge-based, using information measures and explore the benefits of merging them.

**2 Data and methods**

**2.1 Concepts and measures from Information Theory**

Since its beginnings in communication theory and the seminal paper of Claude Shannon (Shannon, 1948), Information Theory (IT) has developed into a scientific discipline of its own, with applications ranging from Meteorology (Brunsell





2010) and Hydrology (Pechlivanidis et al. 2016, Gong et al. 2014, Loritz et al. 2018) to Geology (Wellmann and Regenauer-Lieb 2012) any many others. Information Theory has been proposed as one important approach to advance catchment hydrology to deal with predictions under change (Ehret et al. 2014). A good overview on applications in Environmental and Water Engineering is given in Sing (2013), Cover and Thomas (1991) provide a very accessible yet

comprehensive introduction to the topic.

Please note that while the concepts of IT are universal and apply to both continuous and discrete data and distributions, we will for the sake of clarity and brevity restrict ourselves to the latter case and work with discrete (binned) probability distributions throughout all experiments.

*Information*

The most fundamental quantity of IT, information $I(x)$, is defined as the negative logarithm of the probability $p$ of an event $x$ (see Eq. 1) and depending on the base of the logarithm it is measured in [nat] for base e, [hartley] for base 10, or [bit] for base 2.

$$I(x) = -log_2 p(x) \tag{1}$$

We will stick to unit bits here as it is the most commonly used (especially in the computer sciences) and because it offers some intuitive interpretations. Information in the context of IT has a fundamentally different meaning than in colloquial use, where it is often used synonymously with 'data'.

Information can be described as 'the property of a signal that effects a change in our state of belief about some hypothesis (Nearing et al., 2016, section 2.3). This definition has important implications: Firstly, in order to quantify the information

content of a signal, we have to know (or at least have an estimate) of its occurrence probability *a priori*. Secondly, missing information can be seen as the distance between our current state of belief about something and knowledge, which establishes a link between the concepts of uncertainty and information. Information is carried by data and interestingly, the information content of the same data reaching us can be different depending on our prior state of belief: If we already have knowledge prior to receiving the data, its information content can only be zero and in all other cases it can at most be as

large as our prior state of belief. Information can therefore be interpreted as a measure of surprise: The less probable an event is, the more surprised we are when it occurs, and the more informative are the data revealing this to us. The dependency of information content to our prior state of belief is expressed by the prior probability $p$ we assign to the particular event.

Compared to working with probabilities, using its log-transforms, information, has the welcome effect that counting the total information flown by a sequence of events is additive, which is computationally more convenient than the multiplicative

treatment of probabilities.

*Information Entropy*



Information Entropy $H(X)$, or simply Entropy is defined as the expected or average value of information (see Eq. 2) of a specific value or bin of a data set $X = \{x_1, x_2, \dots, x_n\}$. This means that if information is the additional insight gained from the disclosure of the outcome of a *single* random experiment, then Entropy is the average additional insight if the experiment is repeated *many* times, thus independent, identically distributed (iid) sampling of the entire underlying probability density

function (pdf). Again, we can use the terms expected information and expected uncertainty as flip-side expressions of the same thing (more details on the interpretation of Entropy as uncertainty are presented by Weijs, 2011, chapter 3).

$$H(X) = \boldsymbol{E}[I(x)] = -\sum_{x \in X} p(x) \; log_2 \, p(x) \tag{2}$$

In Eq. 2, $I(x)$ indicates information and $E$ is the expected value.

While information is a function of the occurrence probability of a particular outcome only, Entropy is influenced by and a measure of the shape of the entire pdf. A pdf with only a single possible state of probability $p = 1$ therefore has an Entropy of zero: There will be no surprise from the disclosure of any random draw as we already know the result in advance. If on the other hand the probability mass is spread evenly over the entire value range, Entropy will be maximized. Uniform distributions therefore serve as maximum Entropy (minimum artificially added information) estimates of unknown pdfs.

Like the variance of a distribution, Entropy is a measure of spread, but there are some important differences: While variance takes into account the values of the data and is expressed in (squared) units of the underlying data, Entropy takes into account the probabilities of the data and is measured in bit. Variance is influenced by the relative position of the data on the measure scale and dominated by values far from the mean; Entropy is influenced by the distribution of probability mass and dominated by large probabilities. Some welcome properties of Entropy are that it is applicable to data that cannot be placed

on a measure scale (categorical data), and that it allows comparison of distributions from different data due to its generalized expression in bit.

*Conditional Entropy*

So far we know that Entropy is as a measure of the expected information of a single distribution. We could also refer to this

as a measure of self-information, or information we have about individual data items when knowing the data distribution.

If we do not only have a data set $X = \{x_1, x_2, \dots, x_n\}$ of single data items available, but joint data sets of paired data items $X, Y = \{x_1\,y_1, x_2\,y_2, \dots, x_n y_n\}$, then an obvious question to ask is 'What is the benefit of a priori knowing $Y$ when we want to guess $X$?'. In practice, this situation appears each time we want to make predictions about a quantity of interest (e.g. ground rain rate) by exploiting available, related data (e.g. reflectivity). In the following, we will refer to the former as the target, to

the latter as predictor(s).

Expanding the definition of Entropy, Conditional Entropy $H(X|Y)$ is defined as the probability-weighted (expected) Entropy of all distributions of $X$ conditional on the prior knowledge of $Y = y_i$ (see Eq. 3).





$$H(X|Y) = \sum_{y \in Y} p(y)\, H(X|Y = y) = -\sum_{y \in Y} p(y) \sum_{x \in X} p(x|y)\ log_2\, p(x|y) \tag{3}$$

If $X$ and $Y$ are completely independent, then prior knowledge of y when guessing x from a data pair $(x.y)$ does not help at all. In this case each conditional distribution $p(X|Y = y_i)$ equals the marginal distribution $p(X|Y)$ and the Conditional Entropy $H(X|Y)$ is exactly equal to the unconditional Entropy $H(X)$. While such a situation is clearly not desirable, it also provides us with the useful insight that even if we apply a completely useless predictor, we can never make worse guesses than if we ignored it altogether. This is called the Information Inequality (Eq. 4), or simply 'Information can't hurt'. For the complete proof, see Cover and Thomas (2006) p. 28.

$$H(X) \geq H(X|Y) \tag{4}$$

*Cross Entropy*

Entropy and Conditional Entropy measure information contained in the shape of distributions and the underlying data. Their calculation depends on prior knowledge of the pdf from the data. In practice, this condition is not always fulfilled and we work with estimates of the pdf instead. This means we estimate the information attached to a signal based on imperfect premises, which we pay for with increased uncertainty. Cross Entropy $H_{pq}(p||q)$ as defined in Eq. (5) quantifies exactly this.

$$H_{pq}(p||q) = -\sum_{x \in X} p(x)\, log_2\, q(x) \tag{5}$$

In Eq. (5), $p$ is the true probability distribution of the data, $q$ the assumed-to-be-true distribution. This means we calculate the information content of a particular signal based on it's a priori known, but only approximate occurrence probability $q(x)$. However, as we draw from the real distribution $p$, this particular signal really occurs with probability $p(x)$. The larger the mismatch between $p$ and $q$, the more additional questions we have to ask. In the best case, if premises are correct and $q$ is identical to $p$, $H_{pq}$ will be equal to $H$. In the worst case $H_{pq}$ will go to infinity. An accessible and comprehensive treatment of this topic can be found in Weijs (2011), chapter 3.5.

*Kullback-Leibler divergence*

If both the Entropy of a distribution and the Cross Entropy between the true distribution and a model thereof are known, we can separate these two components of total uncertainty: Uncertainty due to the shape of the true distribution, and uncertainty because we do not know it exactly. The latter is measured by the Kullback-Leibler divergence $D_{KL}(p||q)$ (see Eq. 6 and Kullback and Leibler 1951). It can either be calculated by the probability-weighted difference of true and assumed-to-be-true occurrence probabilities, or by subtracting the Entropy from the Cross Entropy.



$$D_{KL}(p||q) = \sum_{x \in X} p(x) \, log_2 \frac{p(x)}{q(x)} = H_{pq}(p||q) - H(p) \qquad (6)$$

## 2.2 Methods

*Data-based models and predictions, information-based model evaluation*

Suppose we have a data set of many repeated, joint observations of several variates (data tuples), e.g. a time series of joint observations of R and Z. For each data tuple we can consider their values as coordinates in a multi-dimensional space, whose number of dimensions equals the number of variates. Let us further suppose we choose, separately for each variate, a value range and a strategy to subdivide it into a finite set of bins. The first can be based on the observed range of data or on physical considerations, the latter on the resolution of the raw data or on user requirements about the resolution at which the

data should be evaluated or predictions should be made. In this context, many methods for choosing an optimal binning strategy have been suggested (e.g. Knuth 2013), but still the most popular and straightforward method that at the same time introduces minimal side information is uniform binning, i.e. splitting of the value range into a finite number of equal-width bins. Throughout this paper, we will stick to uniform binning.

    With range and binning chosen, we can map the data set into a multivariate, discrete frequency distribution as shown in Fig.

1, left plot. Normalization with the total number of data tuples yields the corresponding multivariate, discrete probability distribution (dpd). Note that the mapping comes at the price of a certain information loss: Firstly, we lose the information about the absolute and relative position of the data tuples in the data set, secondly we lose any information about the data tuple values at resolutions higher than the bin width. Nevertheless, the way how probability mass is spread within the coordinate space is an indication of its structuredness, and its Entropy is a direct measure thereof. High values of Entropy

indicate that probability mass is widespread, which also means we are highly uncertain when guessing a particular data tuple randomly drawn from the dpd. On the other side, if all probability mass is concentrated in a single bin, Entropy will be zero and we can with zero uncertainty predict the values of a data tuple randomly drawn from the dpd.

    So far we used Entropy to measure uncertainty if we'd have to guess all values of a data tuple randomly drawn from an a priori known distribution. But often we do not just know the distribution a priori, but also the values of parts of the data

tuples (e.g. we know Z and the empirical Z-R relation and want to predict R). In this case, our uncertainty about the target variate R given the predictor variate Z and the relation among the two can be measured by Conditional Entropy, which we know from Eq. 4 is always smaller or equal the unconditional Entropy of the target. In Fig. 1, left plot, the conditional frequency distribution of the target $X$ given predictor $Y$ is shown in red, in the right plot, also in red, the related conditional probability distribution and, in blue, the unconditional probability distribution of the target.

In short, we can consider a dpd constructed from a data set as a simple predictive model, which, given some values for the predictor variates provides a probabilistic prediction of the target variate. The advantage of such a model and prediction is



that it provides a joint statement about the target value and the uncertainty associated with that statement, expressed by the spread of the predictive dpd of the target. And as long as we remain with our predictions within the domain of the data we used for constructing the model, our uncertainty statements are accurate, i.e. they are neither over- nor underconfident.

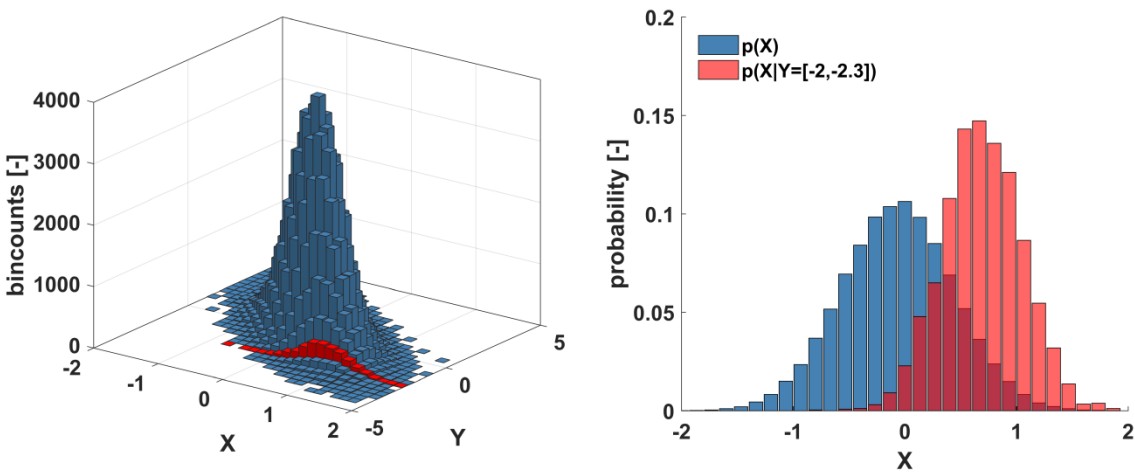

**Fig.1.** Illustration of a bivariate conditional probability distribution as a simple data-based predictive model. Left: Joint histogram of target X and predictor Y (blue). Conditional histogram of target X given predictor values from the interval [-2,-2.3] (red). Right: Marginal (unconditional) probability distribution of target X (blue) and conditional probability distribution of target X given predictor values from the interval [-2,-2.3] (red).

Things are different, though, if we apply the model to a new situation, i.e. if we construct the model from one data set and use it with predictor data of another under the assumption that the predictor-target relation expressed by the model also holds for the new situation. This is clearly only the case if the learning and the application situation are identical, in most cases however, the situations will differ and we will pay for this inconsistency with additional uncertainty. We can measure this uncertainty by Kullback-Leibler divergence (Eq. 6) between the dpd we use as a model and the true dpd of the application

situation. Total predictive uncertainty is then expressed by Cross Entropy (Eq. 5) and can be calculated as the sum of the predictive uncertainty given a perfect model (expressed by Conditional Entropy), which is limited by the information content of the predictors about the target, and the inadequacy of the model (expressed by Kullback-Leibler divergence).

We will apply this approach to construct and analyse predictive models throughout the paper, and use it to learn about the information content of various predictors about our target, ground rainfall R, and learn about the information losses when

applying predictive models that were e.g. constructed from limited data sets, constructed at different places, or simplified by functional compression. Note that expressing relations among predictor and target data by dpd's is not limited to data-based models, in fact functional relationships of any kind (e.g. the Marshall-Palmer Z-R relation) can be expressed in a dpd. In that sense, the framework for model building and testing together with the measures to quantify uncertainty we use here are universally applicable.





*Benchmark models and minimum model requirements*

Expressing predictions by probability distributions, and expressing uncertainty as 'information missing' as described in the previous sections has the advantage that we can build default models providing lower and upper bounds of uncertainty, which we can then use as benchmarks to compare other models against. The smallest possible uncertainty occurs if the

predictive distribution of the target is a Dirac, i.e. the entire probability mass is concentrated in a single bin. In such a case, irrespective of the number of bins covering the value range, the Entropy of the distribution is zero: $H_{Dirac} = 0$. If such a case occurs, we know we have applied fully informative predictors and a fully consistent model. Things are more interesting when we want to formulate upper bounds: The worst case occurs if we use a model which is unable to provide a prediction for a set of predictors in the application case. This happens when the particular situation was never encountered in the

learning data set, but appears during application. In that case, $p(x)$ in Eq. 6 will be non-zero, $q(x)$ will be zero, and Kullback-Leibler divergence will be infinite, indicating that the model is completely inadequate for the application situation. In this case total uncertainty will also be infinite, no matter how informative the predictors are. Infinity as an upper bound for uncertainty is not very helpful, though, and we can do better than that: If the above case occurs, we can argue that the model is inadequate just because it learned from a limited data set, and since we require the model to provide a prediction for all

predictive situations, we can allocate a small but non-zero probability to all bins of the model. In this paper, we did so using the minimally invasive maximum Entropy approach suggested by Darscheid et al. (2018). Note that this was only required in Experiments 2, 3 and 4, when we calculated Kullback-Leibler divergence between a reference and models based either on very small samples (Experiments 2 and 3) or deterministic relations (Experiment 4).

Given that we successfully avoid infinite Kullback-Leibler divergence, the worst thing to happen is that our predictions are

completely uninformative, i.e. we provide a uniform distribution across the entire value range of the target. In this case, the Entropy of the distribution is equal to the logarithm of the number of bins: $H_{uniform} = log_2(number\ of\ bins)$. This means that for all cases where we are sure that infinite Kullback-Leibler divergences will not occur, we can provide an upper limit of total uncertainty which is dependent on the required resolution of our predictions.

Finally, for the special cases where we know that the model we use is perfect (typically because we apply it to the same

situation it was constructed from), we know that Kullback-Leibler divergence is zero, and total uncertainty equals Conditional Entropy. In this case, we can state another upper bound for uncertainty: If at worst the predictors we use are completely uninformative, Conditional Entropy will be equal to the Unconditional Entropy of the target distribution (see Eq. 4).

We will use these lower and upper bounds for total uncertainty throughout the experiments to put the performance of the

tested models into perspective.

*Sampling strategy*

In Experiments 2 and 3 we investigate the effect of limited sample size, i.e. the information loss (or uncertainty increase) if we construct a model not from a full data set, but from subsets drawn thereof. This corresponds to the real-world situation of



building models from available, limited data. For the sake of demonstration we assume in the experiments that a long and representative reference data set is available for evaluation. While this is clearly not the case in real-world situations, we can get answers from such experiments for practically relevant questions such as 'what is a representative sample, i.e. how many observations are required until a model build from the data does not further change when adding more observations?' or

'should we build a model from locally available but limited data, or should we use a model learned elsewhere, but from a large data set?'.

Throughout the experiments, we apply simple random sampling without replacement to take samples from data sets. In order to reduce effects of chance, we repeat each sampling 500 times, calculate the results for each sample and then take the average.

**2.3 Data**

This study uses data from various sources collected in a four-year period (Oct 1, 2012 – Sep 30, 2016) within the research project CAOS (**C**atchments **A**s **O**rganised **S**ystems). For more detail on project goals and partners see Zehe et al. (2014). QPE was an important component of CAOS, and to this end a focus was laid on measuring rain rates and drop size distributions by six laser optical disdrometers, two vertical pointing K-band micro rain radars, standard rain gauges and

weather radar. Table 1 provides an overview on these and additionally used data.

*Study area and hydroclimate*

The project was conducted in the Attert basin, which is located in the central western part of the Grand Duchy of Luxembourg and partially in Eastern Belgium (see Figure 2) with a total catchment area of 288 km². The landscape is

orographically slightly structured, with a small area underlain by sandstones in South, with heights up to 380 m a.s.l, a wide area of sandy marls in the centre part, in which the main Attert river flows from west to east, and an elevated region in the North which is part of the Ardennes massif and reaches elevations up to 539 m a.s.l.

The study area is situated in the temperate oceanic climate zone (Cfb according to the Köppen–Geiger classification, Köppen and Geiger 1930). Precipitation is mainly associated with synoptic flow regimes with a westerly wind component and

amounts to around 850 mm (Pfister et al., 2005 and Pfister et al., 2000), ranging from 760 mm in the centre to about 980 mm in the north-western part (Faber, 1971) due to orography and luv-lee effects. Especially during intensive rain events within a synoptic north-westerly flow regime, the main part of the Attert region lies in the rain shadow of the Ardennes, whereas the outermost north-western part is excluded from this lee effect and still within the region of local rain enhancement by upward orographic lift (Schmithüsen, 1940). This also partly explains local differences in the annual precipitation regime: While in

the elevated regions of the north-west, the main rain season is (early) winter, in the central lowlands convective summer rainfall also provides a substantial contribution. Interestingly, the latter mainly occurs along two main storm tracks: One north-eastern starting in the south-west corner of Luxembourg and touching the study area in the east; one north-eastern starting in Belgium and touching the study area in the west.





In our study, we investigated both the effects of location and season on drop-size distributions and ultimately on QPE.

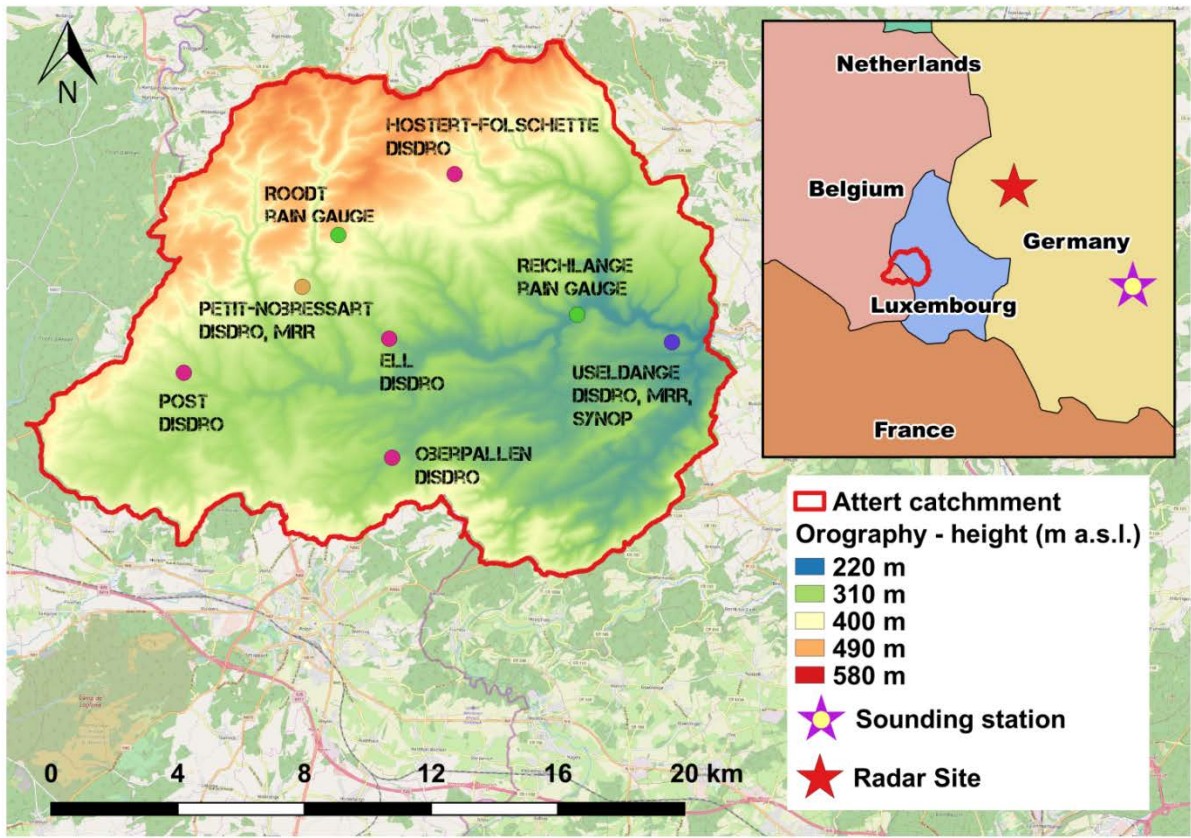

**Fig. 2** The position of the Attert catchment in Luxembourg and Belgium with superimposed Orography (in m a. s.l.) and the locations of the MRR's, disdrometers, rain gauges, the synoptic station (ASTA) and radar site (Neuheilenbach) and the rawinsonde launching
(sounding) station  (Idar-Oberstein- WMO-ID 10618) as well as the scale and  orientation of the small-scale map.

*Radar Data*

We used 10-minute reflectivity data from a single pol C-Band Doppler radar located in Neuheilenbach (see Figure 2), and operated by the German Weather Service (DWD). The raw volume data set has an azimuthal resolution of 1° and 500 m radially, the latter a mean of the 125 m range resolution (short pulse). The antenna's -3dB-beamwidth is 1°. The distance of the study area to the radar site is between 40 and 70 km, which renders high-resolution data and avoids cone of silence issues. The raw data were filtered by static and Doppler clutter filters and bright band correction (Hannesen, 1998) but no
attenuation corrections were applied. Second trip echoes (Bückle, 2010) as well as obvious anaprop echoes were also removed, as especially the latter are prominent in this area during fall and spring (Neuper, 2009). From the corrected data we



constructed a pseudo PPI (plan position indicator) dataset at 1500 m above ground and, to make the data compare- and combinable to all other data used, took hourly averages.

*MRR Data*

We also used drop size spectra measured at 1500 meter and 100 meter above ground by two vertical pointing K-band METEK micro rain radars (MRR) (Löffler-Mang et al. 1999; Peters et al. 2002) located at the sites Useldange and Petit-Nobressart (Figure 2). We operated the MRRs at 100 vertical meters and 10 seconds temporal resolution, but for reasons of storage and processing efficiency did all further processing on 1-minute aggregations thereof. The raw Doppler spectra were transformed to drop size distributions via the drop size – fall velocity relation given in Atlas et al (1973). From the drop size distributions the rain rate and the reflectivity were calculated using the $3.5^{th}$ and the $6^{th}$ statistical moments of the drop size distributions. In doing so we assumed the vertical velocity of the air to be negligible, although this may sometimes play a role (Dotzek and Beheng 2001). As before, we converted the data to 1-hour values by averaging (reflectivity) and summation (rain sum).

*Disdrometer Data*

We also deployed six second generation OTT Particle Size and Velocity (PARSIVEL², Löffler-Mang and Joss, 2000) optical disdrometers in the study area to measure drop size distributions at ground level (Figure 2) in 1-minute resolution. Two were located at the same sites as the MRRs (Useldange and Petit-Nobressart). The others were placed such as to both capture the hydroclimatic variations in the study area and to cover it as uniformly as possible. We applied a quality control to the raw data as described by Friedrich et al. (2013 a, b), converted the filtered data to drop-size concentrations per unit air volume to make them comparable to the weather radar and MRR data, then converted them to reflectivity and rain rate using the $3.5^{th}$ and $6^{th}$ statistical moments of the drop size distributions and finally took 1-hour averages and sums thereof.

*Rain gauge data*

Next to the rain rate retrieved from the disdrometer data, we also used additional observations from standard tipping bucket rain gauges at sites Useldange, Roodt and Reichlange (Figure 2). Quality-controlled rain gauge data from Useldange and Roodt were provided by the Administration des services techniques de l'agriculture (ASTA), who operates a nationwide network of surface weather stations for agricultural guidance. Raw rain gauge data at Reichlange were provided by the Hydrometry Service Luxembourg. We applied plausibility checks to all these data, eliminated questionable data, cases with solid precipitation (based on the output of the hydrometeor classification algorithm of the disdrometers) and finally took 1-hours sums.

*Additional predictors*



In addition to direct observations of precipitation and drop-size spectra, we collected a number of operationally available data to test their value as additional predictors for QPE. We selected standard meteorological in-situ observations such as 2 m temperature, relative humidity, zonal and meridional wind speeds, season indicators such as month and tenner-day of the year as well as synoptic indices such as convective available potential energy (CAPE) and classified circulation pattern. The

latter two are operationally provided by the German weather service. All predictors are listed in Table 1.

The in-situ observations were taken from ASTA station Useldange (Figure 2), and like all other observations underwent additional quality filtering and 1-hour aggregation. The CAPE values are based on rawinsonde data obtained from station Idar-Oberstein (WMO-ID: 10618) located about 90 km east of the study area with a temporal resolution of 6h. The soundings were downloaded from the University of Wyoming homepage (see *data availability)* and checked for

contamination by convection as described in Bunkers et al. (2006). The CAPE values were calculated as surface based CAPE using the virtual temperature (Vasquez 2017). Selecting CAPE as a descriptor for raindrop size distribution followed the reasoning that updraft strength has an influence on the size of the raindrops (Seifert and Beheng 2006). We used a base-10 logarithm transform of CAPE to assure high resolution at lower CAPE values and at the same time sufficient population of the higher classes.

Classified circulation patterns: The German Weather Service provides daily an objective classification of the prevailing circulation pattern over Europe into one of 40 classes (Dittmann et al. 1995). It is based on numerical weather analysis from one of the operational forecast models of DWD at 12 UTC and considers wind direction, cyclonality (high or low pressure influence), and atmospheric humidity. Selecting the circulation as a predictor was based on the perception that it should contain information about both the prevailing precipitation process (stratiform or convective) and about the presence and

origin of air masses, which could influence drop size spectra due to the presence or absence of characteristic cloud condensation nuclei and ice nucleation nuclei.

*Binning choices*

Using all these data in the methods as described in section 2.2 requires binning. Our general binning strategy was to cover

the entire data range, divide it into as few bins as possible to keep bin populations high, but at the same time use enough bins to resolve the shape of the data distributions. We used uniform binning whenever possible, but for strongly skewed data such as reflectivity and rain rate we applied irregular binning. For the latter, we defined the edges of the first bin such as to cover the range from the smallest possible value (0 mm) and 0.2 mm, which is the detection limit of most tipping bucket rain collectors. So essentially, this bin covers all cases of non-zero but irrelevant rain. For the last bin, we set the upper edge such

that it still covered rainfall based on the largest observed reflectivity value transformed to rain rate by the standard Marshall-Palmer Z/R relation ($Z = 200 R^{1.6}$ ), which was 119.9 mm/h. For the bins in between, we increased bin width with rain rate (see footnote to Table 1) to acknowledge both the increasing uncertainty and sparsity of high rainfall observations. With the binning for rain rate fixed, we applied the Marshall-Palmer relation in reverse to get the corresponding bin edges for reflectivity.



*Data filter*

Rainfall is an intermittent process, and quite expectedly most of our 4-year series contained zeroes for rainfall. If we had used this complete data set for analysis, the results would have been dominated by these dry cases, which however was not

5  what we were interested in. We therefore applied a data filter to select only the hydro-meteorologically relevant cases with measurements from all available stations and at least two rain gauges showing rainfall ≥ 0.5 mm/h. Additionally, cases with solid precipitation were excluded using the output from the disdrometers' present weather sensor software. Altogether 11984 data sets passed this 'minimum precipitation' filter, which amounts to almost 17 months of hourly data.

In Figure 3, for the filtered data binned probability distributions of the most important variables are shown.

**Table 1.** Summary of the raw data used in the experiments: Description, summary statistics, binning.

| Symbol | Unit | Value range* | Binning** | Description |
|---|---|---|---|---|
| RR0 | [mm/h] | [0, 86.98] | [ + ] 18 | Rain rate at ground level, measured at 8 sites by disdrometers and rain gauges |
| dBZ1500Rad | [dBZ] | [-99, 56.28] | [ ++ ] 18 | Decibel of reflectivity, measured 1500 m above ground at 8 sites by weather radar |
| dBZ1500MRR | [dBZ] | [-99, 56.28] | [ ++ ] 18 | Decibel of reflectivity, measured 1500 m above ground at 2 sites by MRR |
| dBZ100MRR | [dBZ] | [-99, 56.28] | [ ++ ] 18 | Decibel of reflectivity, measured 100 m above ground at 2 sites by MRR |
| dBZ0Dis | [dBZ] | [-99, 56.28] | [ ++ ] 18 | Decibel of reflectivity, measured at ground level at 6 sites by Disdrometer |
| logCAPE | [log J/kg] | [0, 11.3] | [0:1:12] 13 | Logarithm of the surface based convective available potential energy (CAPE) |
| GWLo | [-] | [1, 40] | [1:1:40] 40 | Circulation pattern according to German Weather Service objective classification |
| TA2 | [°C] | [-15.2, 36.4] | [-20:5:40] 13 | Air temperature, measured 2 m above ground at station Useldange |
| RH2 | [%] | [18.1, 100] | [0:10:100] 11 | Air relative humidity, measured at 2 m above |



| | | | | |
|---|---|---|---|---|
| | | | | ground at station Useldange |
| U10, V10 | [m/s] | [-13.7, 19.1] | [-14:2:20] 18 | Wind velocity in East-West (u) and North-South (v) direction, measured 10 m above ground at station Useldange |
| ToY | [-] | [1, 36] | [1:1:36] 36 | Tenner-day of the year |
| MoY | [-] | [1, 12] | [1:1:12] 12 | Month of the year |
| Statnum | [-] | [1,8] | [1:1:8] 8 | Station number *** |

\*   [min, max]

\*\* [center of first bin, uniform bin width, center of last bin] number of bins

\*\*\* 1: Ell, 2: Hostert-Folschette, 3: Oberpallen, 4: Petit-Nobressart, 5: Post, 6: Useldange, 7: Reichlange, 8: Roodt

\+ Irregular binning with edges [-0.1 0.19 1 2 4 6 8 10 15 20 25 30 40 50 60 70 90 110 130]

5    ++ Irregular binning with edges [-100 11.5 23 27.8 32.6 35.5 37.5 39 41.8 43.8 45.4 46.6 48.6 50.2 51.5 52.5 54.3 55.7 56.8]

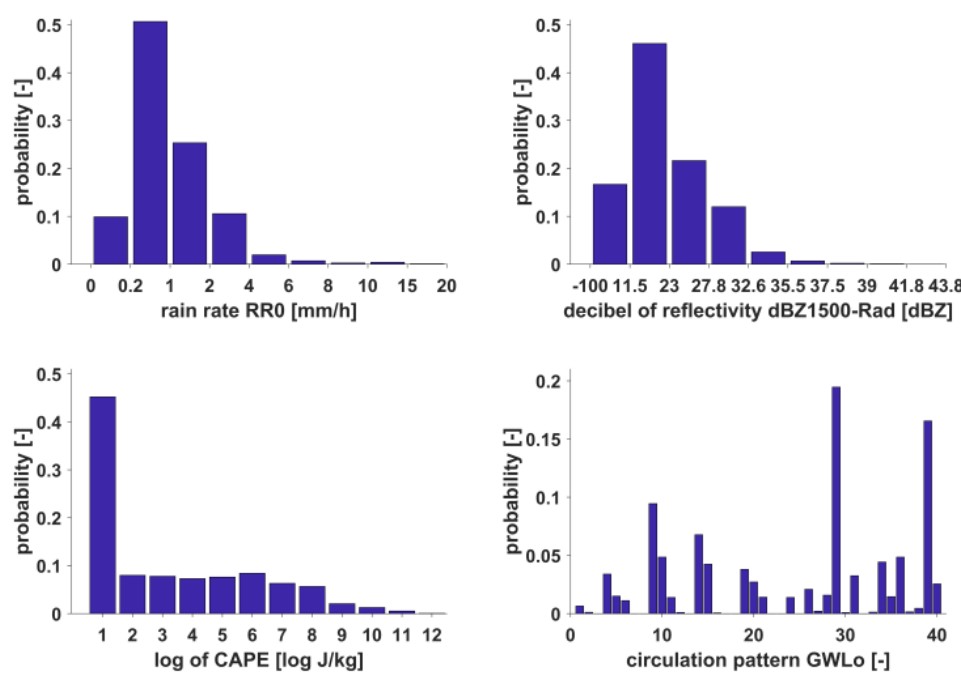

**Fig.3.** Discrete probability distributions of the most important variables used in the experiments (filtered data set). The description of variables and binning can be found in Table 1. X-axes for rain rate and reflectivity are truncated at the right to avoid display of empty bins.



## 3 Results and discussion

### 3.1 Experiment 1: Information in various predictors

In this experiment we explore the information content about ground rainfall R in various predictors (see Table 1). We

selected the predictors with the constraint of being operationally available at any potential point of interest, which applies most importantly to reflectivity measured by weather radar, but also to the predictors we assumed to be spatially invariant within the test domain: convective available potential energy (as surface based CAPE), circulation pattern, air temperature and humidity, wind, and season. We excluded reflectivity measurements by MRR and disdrometer, as these are usually only available at a few locations.

We used ground rainfall observations from the eight rain gauges in the test domain as target data, filtered the raw data with the 'minimum precipitation' filter and created models by various predictor combinations. We measured the usefulness of the predictors with Entropy and Conditional Entropy (Eq. 2 and Eq. 3). The results are shown in Table 2.

**Table 2.** Entropy of the target (RR0) and the benchmark distribution (RR0 uniform), Conditional Entropies of the target given one or

15 several predictors, non-normalized and normalized. Underlying data are from the filtered data set. Predictors are ordered by descending Conditional Entropy.

| case | Predictor(s) | (Conditional) Entropy $H(X)$, $H(X\|Y)$ [bit] | Relative *** (Conditional) Entropy [%] |
|------|--------------|-----------------------------------------------|----------------------------------------|
| 1 | RR0 uniform | 4.17* | 219 |
| 2 | RR0 | 1.90** | 100 |
| 3 | dBZ1500Rad | 1.61 | 84.7 |
| 4 | dBZ1500Rad, RH2 | 1.57 | 82.6 |
| 5 | dBZ1500Rad, U10 | 1.57 | 82.6 |
| 6 | dBZ1500Rad, V10 | 1.57 | 82.6 |
| 7 | dBZ1500Rad, TA2 | 1.55 | 81.6 |
| 8 | dBZ1500Rad, logCAPE | 1.55 | 81.6 |





| 9 | dBZ1500Rad, MoY | 1.54 | 81.1 |
| 10 | dBZ1500Rad, GWLo | 1.52 | 80 |
| 11 | dBZ1500Rad, ToY | 1.47 | 77.4 |
| 12 | dBZ1500Rad, MoY, GWLo | 1.30 | 68.4 |
| 13 | dBZ1500Rad, ToY, GWLo | 1.19 | 62.6 |

\* unconditional Entropy of the benchmark uniform distribution

\*\* unconditional Entropy of the target

\*\*\* H(RR0|predictor(s)) / H(RR0) * 100

There are two upper benchmarks we can use to compare the different QPE models against: Unconditional Entropy if we know nothing but the binning of the target and use a uniform (= maximum Entropy) distribution for prediction (case 1 in Table 2), and the unconditional Entropy of the observed distribution of the target (case 2 in Table 2), which we used as a reference here. The difference between the two is considerable (119%), which means that merely knowing the true distribution of RR0 is already a valuable source of information.

The next important source of information are radar reflectivities: If we use it as a single predictor (case 3), uncertainty is reduced by 0.28 bit or 15.3% compared to RR0. Note that this approach directly applies the reflectivity data provided by the radar, no side information was added nor existing information in the data destroyed by applying an additional Z-R relation. Using each of the other predictors separately did not reduce uncertainty much (not shown), we therefore only show results for the cases where they were applied as 2-predictor models in combination with radar reflectivity (cases 4 to 11). If we

compare the relative Conditional Entropies of their predictions to those of the radar-only model, we can see that neither the ground meteorological observations nor CAPE contained much additional information (cases 4 to 8). Instead, the three most informative models (cases 9 to 11) either distinguish the relation between R and Z by circulation pattern or by season, which corresponds to the operational practice of many weather services to use one Z-R relation for summer and one for winter. Based on these results, we built and evaluated 3-predictor models only with combinations of these relatively informative

predictors (cases 12 and 13). The information gain from using three predictors in combination is considerable, and altogether uncertainty reductions down to 68.4 % (case 12) and 62.6 % (case 13) compared to the 'target-distribution-only case' (case 2) were achieved. The benefit of applying a season- and circulation pattern dependent relation between R and Z also becomes obvious if we compare case 3 (the radar-only model) to cases 12 and 13: In the first case, uncertainty is reduced by 16.3 %, in the latter by 22.1 %.

An obvious conclusion from these findings would be to build better models by simply adding more predictors, which according to the Information Inequality (Eq. 4) never hurts. In fact, when learning from limited data sets, adding enough predictors will in the end result in perfect models with zero predictive uncertainty. However, there is a catch in this known as



overfitting or 'Curse of Dimensionality' (Bellman 1957): In order for a model to be robust in the sense of 'being only weakly sensitive to the presence or absence of particular observations in the learning data set', it must be supported by an adequate number of learning data, and this number grows exponentially with the number of predictors included in the model. So instead of adding more predictors, we will explore the robustness of our models in the next experiment.

## 3.2 Experiment 2: The effect of sample size

The data base and filter used here are identical to the previous experiment, so a set of overall 11984 joint observations of the target (rain rate at the ground) and predictors (radar reflectivity, circulation pattern, tenner-day and month of the year) were available. From these data we built and tested altogether four predictive models as described in Experiment 1: A single predictor model applying radar reflectivity only, two double predictor models applying radar reflectivity and tenner-day or month of the year, and a triple predictor model using radar reflectivity, month of the year and circulation pattern. The difference to Experiment 1 is that now we do not only apply the entire data set but also randomly drawn samples thereof to build the model (see section 2.2 for an explanation of the sampling strategy). Each model is then applied to and evaluated against the full data set. In this case, total predictive uncertainty is measured by Cross Entropy (Eq. 5), which is the sum of Conditional Entropy of the target given the predictors for the full data set and Kullback-Leibler divergence of the sample-based model and the model built from the full data set (see section 2.2).

The results are shown in Fig. 4 as a function of sample size. As in Experiment 1, we included the benchmark uncertainties for applying a maximum Entropy model (red horizontal line, case 1 in Table 2) and a zero-predictor model (black horizontal line, case 2 in Table 2) to put the other models into perspective.





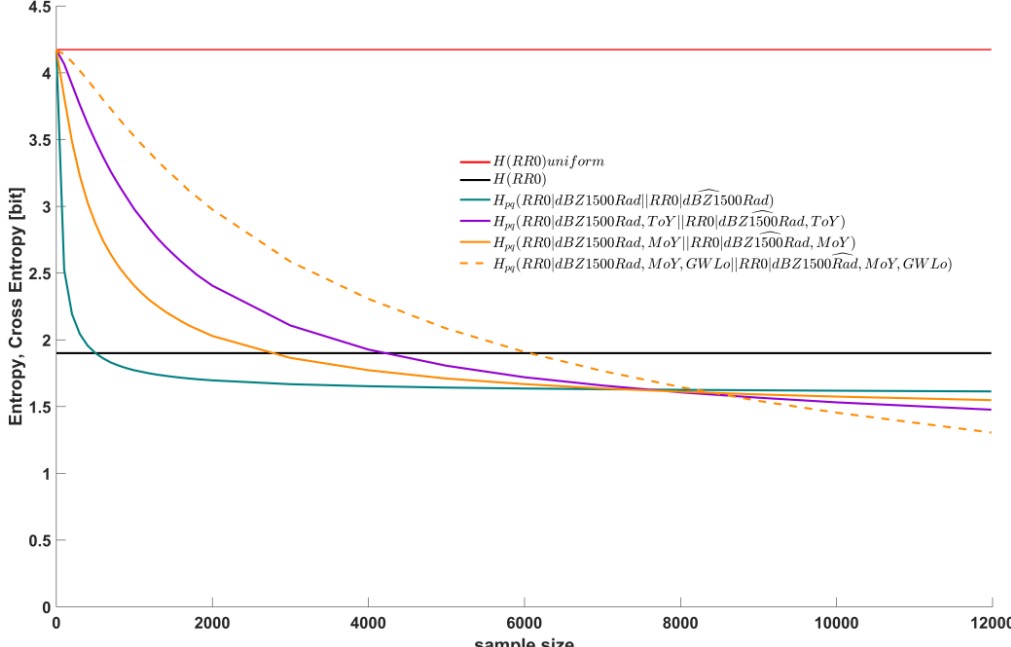

**Fig. 4.** Entropy of the unconditional target distribution (filtered data set, black line), Entropy of the benchmark uniform distribution (filtered data set, red line), Cross Entropies between conditional distributions of the target given one, two and three predictors of the filtered data set and samples thereof (green, purple, yellow and dashed yellow lines). The 'hat' symbol indicates a sample.

At the right margin of Figure 4, Cross Entropies are shown for the case when the sample comprises the entire data set. In this case, Kullback-Leibler divergence is zero, and total uncertainty equals Conditional Entropy. This is the same situation as in Experiment 1 and the values correspond to those in Table 2 for cases 3, 9, 10 and 12. The triple-predictor model (dashed yellow line) quite expectedly outperforms the double predictor models (solid purple and yellow lines), which in turn

outperform the single-predictor model (solid green line). Things look differently, though, if only samples are used for learning. Consider the single-predictor model: For very small sample sizes close to zero, the information content of the sample is close to zero, and hence predictive uncertainty is close to that of the ignorant maximum Entropy model (red line) and considerably higher than that of the zero-predictor model. However, when increasing the size of the sample just a little, its information content quickly rises and Cross Entropy drops. In fact, when learning the relation between radar reflectivity

and ground rainfall from only about 2000 joint observations, the model is almost identical to a model learned from the full data set of 11984 joint observations: Kullback-Leibler divergence is almost zero and Cross Entropy is almost as low as for the model built from the full data set. From this we can conclude that the full data set contains considerable redundancy, which in turn implies that we can build robust single-predictor models from the available data.

We can interpret the lines in Fig. 4 as learning curves, or more specifically they represent the information about the target

contained in samples of different sizes. If we now consider the two double-predictor models and the triple-predictor model and compare it to the single-predictor model, we can see that the more predictors we add, the slower the learning rates





become and the longer learning takes (the curve inclinations are lower for small sample sizes, but remain non-zero for larger samples). For the two double-predictor models this means that samples larger than about 8000 (about two thirds of the data set) are required before their total predictive uncertainty fall below that of the single-predictor model. For the triple-predictor model, even samples larger than about 8500 are required, and interestingly the model continues to learn even for very large

sample sizes (the yellow dashed line is still inclined even for large samples).

The learning behaviour of the models, which differs with the number of predictors used, is a manifestation of the Curse of Dimensionality, and visual examination of learning curves of different models as plotted in Fig. 4 allows two choices: For a given sample size, we can choose the best (= least uncertain) model; for a given size data set, we can choose the model with the best trade-off between performance and robustness. For the latter choice, we can establish selection criteria such as 'a

model qualifies as robust if it learns from at most two thirds of the available data at least 95% of what it can learn from the entire data set' and then choose the best model satisfying this criterion. From the models displayed here, according to our subjective choice, the double-predictor model using a month-specific relation between radar reflectivity and ground rainfall provides the best trade-off between performance and robustness.

### 3.3 Experiment 3: Site-specific Z-R relations

In this experiment we investigate the information content in spatial position by learning and applying site-specific relations between radar reflectivity and ground rain rate (in the previous experiments we applied them in a spatially pooled manner). We used data filtered with the the 'minimum precipitation' filter again, so a set of overall 11984 joint observations of ground rain rate and radar reflectivity were available. We included spatial information by simply using the ID number of each station ('Statnum' in Table 1) as an additional predictor, which means that we built and applied relations between reflectivity

and rain rate at the ground specifically for each station in the test domain (see Fig. 2). As in the previous experiment we evaluated the predictive performance of these models as a function of sample size with Cross Entropy. The results are shown in Fig. 5.





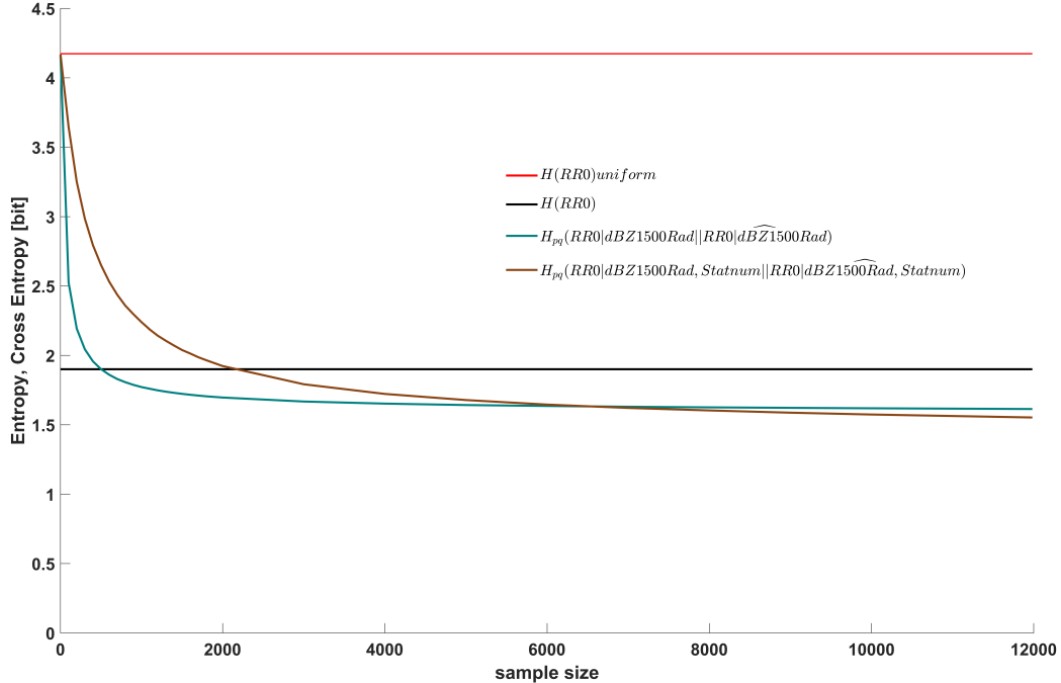

**Fig. 5.** Entropy of the unconditional, target distribution (filtered data set, black line, same as in Figure 4), Entropy of the benchmark uniform distribution (filtered data set, red line, same as in Figure 4), Cross Entropies between conditional distributions of the target of the filtered data set and samples thereof: Single predictor model applying dBZ1500Rad (green line, same as in Figure 4), double predictor model applying dBZ1500Rad & station number (brown line). The 'hat' symbol indicates a sample.

The red and black horizontal lines are the same as in Fig. 4 and as before represent the benchmark unconditional Entropies of the target for a maximum Entropy uniform distribution and the observed distribution. The green line was also included in Fig. 4: It represents the Conditional Entropy of the single-predictor model applying radar reflectivity only. The brown line

shows the performance of the two-predictor model including station ID. The overall pattern is similar to Experiment 2: Adding a predictor reduces total uncertainty if the full data set is used for learning (at the right margin, the brown line lies below the green), but higher-predictor models require more data for learning (the green line descents slower and longer than the brown). Here, the double predictor model outperforms the zero-predictor and the single-predictor model only for samples larger than 2200 and 6500, respectively.

Overall, the information gain of using site-specific Z-R relations is moderate (Cross Entropy of the radar-only model for the full data set is 1.61 bit, and 1.55 bit for the site-specific model) and lower than when distinguishing Z-R relations according to time or circulation pattern (cases 9, 10 and 11 in Table 2). This is not very surprising if we consider the extent of the test domain: The largest distance between two stations is 9.5 km (between stations Oberpallen and Useldange), and the largest elevation difference is 182 m (between stations Reichlange and Roodt). Across these relatively small distances, it appears

reasonable that Z-R relations do not differ substantially. However, this could be different when working in larger domains or





in domains with hydro-meteorologically distinctly different subdomains, such as lowlands and mountain areas or different climate zones (as shown for example in Diem 1968).

### 3.4 Experiment 4: The effect of functional compression

In this experiment we evaluate the effect of functional compression by measuring the information loss when using a deterministic function to express the relation between radar reflectivity and ground rain rate instead of the empirical relation derived from the data. As before, we used all joint observations of radar reflectivity and ground rainfall passing the 'minimum precipitation' filter. Each data pair is shown as a blue dot in Fig. 6, and we can see that there exists a strong, positive and non-linear relation between them. We already made use of this relation in Experiment 1 when we built a single-

predictor model using reflectivity to estimate ground rainfall. Comparing cases 2 and 3 in Table 2 we can see that prior knowledge of reflectivity indeed contains valuable information, reducing total uncertainty by 15.3 % (100 % to 84.7 %).

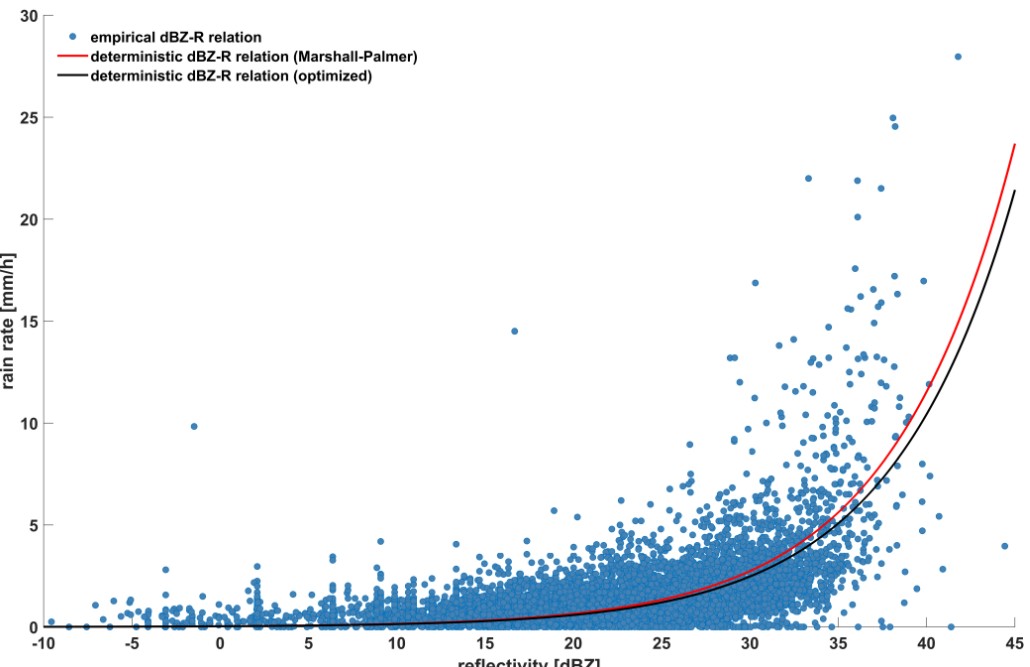

**Fig. 6.** Relation between target RR0 and predictor dBZ1500Rad. Empirical relation as given by the filtered data set (blue dots).
Deterministic power-law relation according to Marshall and Palmer, 1948 (red line); deterministic power-law relation with optimized parameters a = 235 and b = 1.6 (black line).

Let us suppose we would not have been in the comfortable situation to have joint observations of target and predictor to construct a data-based model, or suppose it would take too much storage or computational resources to either store or apply such a model. In such cases, it could be reasonable to approximate the 'scattered' relation as contained in the data either by a



deterministic function gained from other data or a deterministic function fitted to the data. In fact this is standard practice. Expressing a data-relation by a function drastically reduces storage space, is easy to apply, and preserves the overall relation among the data. However, what we lose is information about the strength of that relation as expressed by the scatter of the data. Instead, when applying a deterministic function we claim that the predictive uncertainty of the target is zero.

Our aim here is to quantify the information loss associated with such deterministic functional compression. Applying a deterministic model is in principle no different than using a model learned from a subset of the data as we did in Experiment 2: We use an imperfect model, and this will result in *additional* uncertainty which we can measure via Kullback-Leibler divergence between the true data relation and the model we apply (see Eq. 6). Total predictive uncertainty is then measured by Cross Entropy (Eq. 5) as the sum of Conditional Entropy (the uncertainty due to incomplete information of the predictor

about the target) and Kullback-Leibler divergence (see section 2.2).

For demonstration purposes we applied two typical deterministic Z-R relations of the form $Z = a \cdot R^b$. The first is the widely used Marshall-Palmer relation (mostly attributed to Marshall and Palmer, 1948 as for example in Battan 1959b and Sauvageot 1992, but where this particular formula actually comes from is not known to the authors) with parameters $a = 200$ and $b = 1.6$. The Marshall-Palmer relation is often used as a default model if no better options or local data are available. For

the second, we assumed the local reflectivity and rain rate observations to be available and used them to optimize parameter $a$ by minimizing the root mean square error (RMSE) between the observed rain rate and the modelled rain rate using reflectivity observations. Following the recommendations by Hagen (2003) we only varied $a$ and kept $b$ constant at 1.6. In the end, the optimized $a = 235$ was not far from its Marshall-Palmer pendant and reduced RMSE only slightly from 1.18 mm/h (Marshall-Palmer relation) to 1.17 mm/h. Apparently, the default Marshall-Palmer relation already nicely fit our data.

In Fig. 6, the two deterministic Z-R functions are plotted as red (Marshall-Palmer) and black (optimized Z-R relation) lines. Both capture the overall shape of the empirical Z-R relation quite well, except for high reflectivities where they tend to underestimate the observed rain rates. Applying such a deterministic Z-R relation to a given reflectivity observation is straightforward and yields a prediction of the related rain rate. However, from such a single-valued prediction we cannot infer the related predictive uncertainty, and the best we can do is to additionally provide the model's RMSE as a proxy.

As described above, we additionally used our information-based approach and calculated the conditional Kullback-Leibler divergence between the predictive distributions given by the empirical (= perfect) and the deterministic (= imperfect) models for all available data. From Eq. 6 we see that when comparing a reference distribution $p$ to a model distribution $q$, a situation can occur where the latter is zero but the former is not, which means that an event is contained in the reference data that, according to the model, can never occur. In such a case Kullback-Leibler divergence will be infinite, branding the model as

completely inadequate. However, this verdict may seem too hard, e.g. if the model shows otherwise good agreement with reality, if we have reason to believe that the mismatch only occurred from a lack of opportunity due to a small data set rather than due to a principal mismatch of model and reference or, as in our case, if we know that we use a deterministic approximation instead of the real data relation. In such cases, infinite divergence can be avoided by padding empty bins of





the model distribution with small but non-zero probabilities, which we did by applying the minimally invasive maximum Entropy approach suggested by Darscheid et al. (2018).

For our data, Conditional Kullback-Leibler divergence for the Marshall-Palmer model was 3.43 bit, and 2.69 bit for the optimized model. Added to the Conditional Entropy of the empirical Z-R relation, this resulted in total predictive uncertainties of 5.04 and 4.30 bit, respectively. In terms of relative contributions, this means that 68% (Marshall-Palmer) and 62.5% (optimized model) of total uncertainty are due to deterministic functional compression. This is quite considerable, and even more so if we compare these results to the two benchmark cases in Table 2 (cases 1 and 2): Even the default and safe-side model of applying a uniform distribution (case 1) involves smaller predictive uncertainty than the deterministic models. This seems counter-intuitive at first when recalling the good visual agreement of the empirical and deterministic Z-R relations in Fig. 6. The reason for such large Kullback-Leibler divergences are the relatively high binning resolution for rain rate (Table 1) in combination with the way Kullback-Leibler divergence is computed: Probability differences are calculated bin by bin and irrespective of probabilities in neighbour bins. This means that even a small over- or underestimation of a model, with an offset of the main probability mass by just one bin compared to observations results in large divergence, which can in the end even exceed that of a prudent model spreading probability mass evenly over the data range. For the data used in this experiment, we assume that the agreement between deterministic model predictions and observations would quickly increase when coarse-graining the binning. This would be an interesting question to pursue in a future study, for now we restrict ourselves to the main conclusion of this experiment: As long as learning about and application of data-relations is carried out in the same data set, compression of probabilistic data-relationships to deterministic functions will invariably increase uncertainty about the target. However for the forward case, i.e. cases where there are no data available for learning but predictions are nevertheless required, application of robust deterministic relations capturing the essential relation between available predictors and the target is nevertheless useful.

## 3.5 Experiment 5: Information gains along the radar path

In this experiment we explore how the information content about ground rainfall in reflectivity observations is related to the measurement position along a vertical profile above the rain gauge, and how it is related to the measurement device. To this end, we used data from two sites, Petit-Nobressart and Useldange (see Fig. 2), where a range of reflectivity observations along a 1500 m vertical profile starting at ground level was available: disdrometer observations of reflectivity and rain rate at ground level, MRR reflectivity observations in 100 meter resolution between 100 and 1500 m above ground, and observations from C-band weather radar at 1500 m. With the goal in mind to provide guidance for the layout of future QPE sensor networks we addressed the following questions: 'Are MRR observations at 1500 m more informative than weather radar observations taken at the same height?'; 'How much information is gained if we use near-surface instead of elevated MRR observations, thus omitting the influence of the vertical profile of reflectivity?'; and finally 'How much information is lost if we measure reflectivity instead of rainfall at ground level?'.



Differently from the previous experiments, we were restricted to two (instead of eight) sites with MRR's installed. This made application of the standard 'minimum precipitation' filter (see section 2.3) inappropriate. So instead of filtering the raw data by at least two rain gauges showing rainfall ≥ 0.5 mm/h, we now applied this threshold separately to each of the two sites. At Petit-Nobressart altogether 1241 data tuples passed the filter, at Useldange 612 (Useldange was put into operation a year

5     after Petit-Nobressart).

As in Experiment 1, we used Entropy and Conditional Entropy to measure the information content of the available predictors and added the Entropy of both a uniform and the observed distribution of ground rainfall as benchmarks. The results are shown in Table 3. As the results for the two stations are similar, we mainly discuss Petit-Nobressart in the following, moving from the remotest and presumably least informative predictor to the closest.

**Table 3.** Separately for sites Petit-Nobressart and Useldange: Entropy of the target (RR0) and the benchmark distribution (RR0 uniform), Conditional Entropies of the target given various predictors along the radar path, non-normalized and normalized. Underlying data are from the filtered data set. Predictors are ordered by decreasing distance to the target.

| | | Location Petit-Nobressart | | Location Useldange | |
|---|---|---|---|---|---|
| case | Predictor(s) | (Conditional) Entropy H(X), H(X\|Y) [bit] | Relative *** (Conditional) Entropy [%] | (Conditional) Entropy H(X), H(X\|Y) [bit] | Relative *** (Conditional) Entropy [%] |
| 1 | RR0 uniform | 4.17* | 242 | 4.17* | 250 |
| 2 | RR0 | 1.72** | 100 | 1.67** | 100 |
| 3 | dBZ1500Rad | 1.45 | 84.3 | 1.41 | 84.4 |
| 4 | dBZ1500MRR | 1.48 | 86.0 | 1.37 | 82.0 |
| 5 | dBZ100MRR | 1.27 | 73.8 | 1.30 | 77.8 |
| 6 | dBZ0Dis | 1.14 | 66.3 | 1.14 | 68.3 |

15     * unconditional Entropy of the benchmark uniform distribution

** unconditional Entropy of the target

*** H(RR0|predictor) / H(RR0) * 100

Weather radar data, even if they are measured at far range and at height contain considerable information about ground

20     rainfall: Using them as predictors reduced uncertainty by 15.7 % (100% to 84.3%) compared to the benchmark Entropy of



the target distribution (Table 3, cases 2 and 3). This is comparable to the outcomes of Experiment 1 based on all sites (15.3%, Table 2, same cases). We expected MRR observations taken at the same site and elevation to be more informative than their weather radar pendant, because the MRR signal path and with it the potential for signal corruption is considerably shorter. This is however not clearly evident from the results: At Useldange, using MRR instead of weather radar data

additionally reduced uncertainty by only 2.4% (84.4% to 82.0%, Table 3, cases 3 and 4), and at Petit-Nobressart they were even less informative (86.0% instead of 84.4%, Table 3, same cases). These results should be interpreted with some care due to the relatively limited data base, however it seems safe to conclude that there is no large difference in the information content of MRR and weather radar observations taken at height.

So why take the extra trouble to operate an MRR? Its benefit shows when moving down to elevations inaccessible by

weather radar. Changes in drop size distribution along the pathway of rainfall from cloud to ground can be considerable, and the closer to the ground the observation is taken the stronger its relation to ground rainfall: Using MRR data from the downmost bin ($\leq$ 100 m above ground) reduced uncertainty by 12.2% compared to using the uppermost bin (86.0% to 73.8%, Table 3, cases 4 and 5). This underlines the importance of VPR correction when using weather radar data (Vignal et al. 1999, Vignal et al. 2000). Based on these results, an obvious next step would be to derive VPR corrections from the two

MRR's and investigate the information gain when applied to other sites in the domain; however for the sake of brevity we leave this for future studies.

Further information gains can be achieved when measuring reflectivity directly at the ground: Using the disdrometer measurements further reduced uncertainty by 7.5% (73.8% to 66.3%, Table 3, cases 5 and 6), compared to the MRR observations. This gain is not only due to using ground observations, which completely excludes any negative VPR effects,

but also because predictor and target data were observed at the same spot and by the same sensor. Seen from this perspective, it is even surprising that nevertheless considerable uncertainty remains (66.3% of the benchmark uncertainty, Table 3, cases 6 and 2) which must be attributed to the ambiguous relationship between radar reflectivity and rain rate as measured by the disdrometer.

### 3.6 Experiment 6: QPE based on Radar and rain gauge data

In this final experiment we compare two methods of QPE, radar-based and rain gauge-based and additionally explore the benefits of jointly tapping both sources of information. As we were not dependent on the availability of MRR data as in the previous experiment, we could again make use of the full eight-site set of observations filtered with the 'minimum precipitation' filter.

We used the same data-based approach of constructing empirical dpd's as predictive model like in all previous experiments;

the only difference between the three tested QPE models was the type of predictor used: For the radar-based QPE, we used weather radar observations at 1500 m above ground in a single-predictor model to predict ground rainfall at the same site. This is the same approach as applied in Experiment 1. For the rain gauge-based QPE, we built a single-predictor model based on a straightforward approach comparable to leave-one-out cross validation: Each of the eight available stations (see





Fig. 2 and Table 1) was once used as a target station, and observations from each of the remaining seven stations were separately used as predictors. This way we could calculate the information content in the predictor as a function of the distance between stations. Eight stations render altogether 56 unique station pairings, for our stations the minimum, average, and maximum distances were 1.9, 7.2, and 15.3 km, respectively. For later plotting, we binned the results in seven distance

classes of two km width each and took averages within each bin. This is comparable to using range bins when calculating a semivariogram.

For the joint QPE model we extended the approach used for rain gauge interpolation to a double-predictor model: Again each of the eight available stations was once used as a target station, but now not only observations from each of the remaining stations were (separately) used as predictors, but additionally radar observations measured at height above the

target station. This is a typical approach when merging radar and rain gauge data for QPE: We use data from rain gauges observed at a horizontal, and radar data observed at a vertical distance from the target.

As we build and compared in this experiment models with different numbers of predictors, and as the models for each particular target station were built from a subset of the data only, it could be worthwhile to explore the additional uncertainty due to the effect of sample size here as in Experiments 2 and 3. However, for this particular experiment we found it more

useful for the reader (and us) to discuss results as a function of distance between stations, as it provides a link to the large body of literature on spatial rainfall structure analysis and station-based rainfall interpolation. The results are shown in Fig. 7.

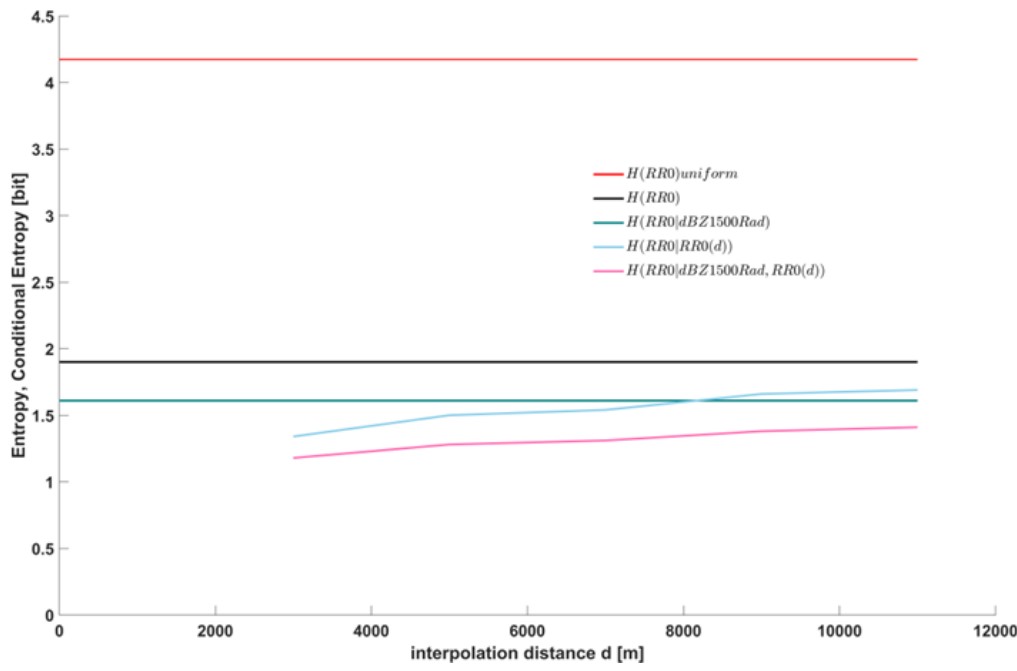





**Fig. 7.** Entropy of the unconditional, target distribution (filtered data set, black line, same as in Figure 4), Entropy of the benchmark uniform distribution (filtered data set, red line, same as in Figure 4), Conditional Entropy of the target given reflectivity as predictor (green line), Conditional Entropy of the target given station rain rate observations as a function of interpolation distance (light blue line), Conditional Entropy of the target given reflectivity and rain rate at stations as predictor, as a function of interpolation distance (pink line).
Underlying data are from the filtered data set.

Just as in Fig. 4 and Fig. 5, we included the benchmark unconditional Entropies of the target for a maximum Entropy uniform distribution (red line) and the observed distribution of all stations put together (black line) to put the results into perspective. As the radar-only model (green line) is independent of any interpolation distance, simply because it does not

make use of any station data, it plots as a horizontal line. Its conditional Entropy (1.61 bit) corresponds to the right-end (= full-size sample) value of the green line in Fig. 4, and to case 3 in Table 2.

For the QPE model based on rain gauge observations (light blue line), interpolation distance however does play a role: As to be expected, the smaller the distance between the target and the predictor station, the higher the information content of the predictor and the smaller Conditional Entropy, i.e. the blue lines rises from left to right. For small distances between two and

four km, Conditional Entropy is 1.34 bit, which is lower than for the radar-only QPE. If we take the unconditional Entropy of the target again as a reference as in Experiment 1 (Table 1, case 2), station-only QPE for short distances reduces uncertainty to 70.5%, radar-only to 84.7%. For large distances however this order is reversed, and station-only QPE reduces uncertainty only to 1.69 bit or 88.9% of the reference. The break-even point at which both methods perform equally well lies at a station distance of about 8 km. This means that if we were asked to choose one of the two models for QPE, the best

choice would be to use station interpolation for all targets within less than about 8 km from the nearest station, and radar-QPE for all other.

The above results indicate that each of the two QPE methods has its particular strengths. In other words they add non-redundant information, and therefore we can expect some benefits when joining them in a two-predictor model. This is indeed the case if we take a look at the related Conditional Entropies plotted as pink line in Fig. 7: While it resembles the

plot of the station-only model in overall shape and trend, it is shifted downward: Conditional Entropy for small distances is now reduced to only 1.18 bit (62.1% of the reference), and even for the largest distances (1.41 bit = 74.2% of the reference) it is still smaller than both the station-only and the radar-only values. Interestingly, in the combined model the radar observations add information even for small station distances. We assume that this can be attributed to convective rainfall situations, where strong spatial rainfall heterogeneity occurs, unresolvable even by rain gauges only a few kilometres apart.

In Experiment 1 we also built and tested two-predictor models. The best among them, using radar data and tenner-days of the year for predictors (Table 1, case 11), reduced Conditional Entropy to 77.4% of the reference. While this is a considerable improvement, it is surpassed by the radar-rain gauge model even for the largest station distance. Obviously, ground rainfall and radar observation contain non-redundant information which, used together, have a large potential to improve QPE.



## 4 Summary and conclusions

Reliable QPE is an important prerequisite for many hydro-meteorological design and management purposes. In this context we pursued two aims with this paper: The first was to suggest and demonstrate a probabilistic framework based on concepts of Information Theory, in which predictive relations are expressed by empirical discrete probability distributions directly derived from data. The framework allows integration of any kind of data deemed useful and explicitly acknowledges the uncertain nature of QPE. The second aim was to investigate the information gains and losses associated with various data and practices typically applied in QPE. For this purpose we conducted altogether six experiments using a comprehensive set of data comprising four years of hourly-aggregated observations from weather radar, vertical radar (MRR), disdrometers, rain gauges, and a range of operationally available hydro-meteorological observables such as large-scale Circulation Patterns, ground meteorological variables and season indicators.

In *Experiment 1*, we measured the information about ground rainfall contained in various operationally available predictors with Entropy and Conditional Entropy. Weather radar proved to be the single-most important source of information, which could be further improved by distinguishing Z-R relationships by season and prevailing circulation pattern.

In *Experiment 2*, we tested the robustness of QPE models developed in the previous experiment by measuring the additional uncertainty due to limited learning data with Kullback-Leibler divergence and Cross Entropy. The main lesson learned here was that this added uncertainty is strongly dependent on the number of predictors used in the model, and that for unfavourable constellations (multiple-predictor models learning from small samples) this effect quickly dominates total uncertainty: This is the well-known Curse of Dimensionality. For the data set used in this study, we found a double-predictor model using a month-specific relation between radar reflectivity and ground rainfall to provide the best trade-off between performance and robustness.

In *Experiment 3*, we examined to which degree the empirical Z-R relationship in the test domain varies in space, and what size data sets are required to support the use of site-specific Z-R relations. Overall, the information gain of site-specific Z-R relations was smaller than distinguishing them according to season or circulation pattern as we did in the previous experiments. Moreover, applying site-specific Z-R relations instead of a single, site-independent one means adding another predictor to the QPE model, which in turn mean that we increase the Curse of Dimensionality. In fact, more than half of the available data had to be used for learning the site-specific model before the additional effort paid off. The conclusion here was to stick to a site-independent relation, which also seems reasonable from a meteorological point of view due to the relatively small extent of the test domain (288km²).

In *Experiment 4*, we evaluated the effect of functional compression, i.e. the information loss when replacing the data-inherent probabilistic relationship between reflectivity and rain rate by a deterministic functional approximation. We measured the additional uncertainty with Kullback-Leibler divergence, which adds to the uncertainty caused by the incomplete information of the predictor about the target measured by Conditional Entropy. We used two standard power-law Z-R relations: The well-known Marshall-Palmer relation, and a modified version thereof with optimized function





parameters. For both, the added uncertainty was in the order of 60% of total uncertainty. While these numbers may vary with the chosen binning resolution, compression of probabilistic data-relationships to deterministic functions will invariably increase uncertainty as long as learning and applying of data-relations is carried out in the same data set.

In *Experiment 5*, we investigated information gains along the radar path using weather radar, MRR and disdrometer data from two sites. The main insights here were that the information content of radar reflectivity measured at height (1500 m above ground) does not differ much between weather radar and MRR, but considerable additional information is gained by using observations from lower elevations (100 m above ground), thus avoiding information losses by changes in drop size spectra along the pathway of precipitation from cloud to ground. This underlines both the importance of VPR corrections for accurate QPE and of the required MRR observations. Despite these information gains, considerable uncertainty remained even when using as predictor reflectivity observations taken at the same spot and by the same device (disdrometer) as the target variable (rain rate) itself. This indicates a somewhat ambiguous relationship between radar reflectivity and rain rate as measured by the disdrometer, which could potentially be sharpened by taking into account precipitation type.

In the last *Experiment 6*, we built QPE models based on radar data only, rain gauge data only and a combination thereof and evaluated their information content as a function of the distance between the target and predictor rain gauge. Comparing the first two revealed that a separation distance of ~ 8 km marks a transition: For smaller distances, gauge-based QPE is superior, for larger distances radar-based QPE. Combining the two in a double-predictor QPE model however not only outperformed both of them for all separation distances, but also competing radar-based double-predictor models applying season or circulation pattern. For rain gauge networks comparable in coverage and density to the one used in this study, rain gauge and radar data apparently contain useful, non-redundant information which should be jointly exploited.

We would like to emphasize that the results from these experiments are partly contingent on the choice of the data filter: The 'minimum precipitation' (at least two rain gauges with rainfall ≥ 0.5 mm/h) filter we applied excluded all cases where rainfall occurred at only a single station, and it also excluded all cases of widespread but very low-intensive rain.

Quantitative statements about the information content of particular predictors or the relative performance of competing QPE models may differ according to these user choices. However, we would like to point out that all of the different experiments in this study could be formulated, conducted and evaluated in a single framework and relied on a single property, information. It is therefore not so much the particular results we want to emphasize here, but rather the probabilistic, data- and information based framework we applied. By its probabilistic concept, it explicitly acknowledges the uncertain nature of QPE, by expressing probabilities in terms of information it facilitates both interpretation and computation. In the framework, predictive relationships are directly derived from data and expressed as discrete probability distributions. The advantage is that this avoids as much as possible introduction of unwanted side information, e.g. by parametric choices, and that it avoids deletion of existing information e.g. by data transformation or lossy compression. Altogether, this facilitates tracking sources and sinks of information. These advantages however come at a price: Learning robust data-based relations requires a considerable amount of available target and predictor data, and applying them for predictions is computationally more expensive than using deterministic functions.





*Data availability:* The reflectivity and rain rate data measured by the six disdrometers and two MRR's is published via the GFZ Data Service repository (Neuper and Ehret 2018). The rain gauge data of the station at Roodt and the rain gauge data, the 2 m temperature data, the relative humidity, and the wind data of the station at Useldange is publicly available

from the Administration des services techniques de l'agriculture ASTA at http://www.agrimeteo.lu/ (last access: 04. December 2018). The rain gauge data of the station at Reichlange was provided upon request by the Hydrometry Service Luxembourg (https://www.inondations.lu/ ;  last access: 04. December 2018). The Sounding Data from which CAPE was calculated is publicly available from the University of Wyoming at http://weather.uwyo.edu/upperair/sounding.html (last access: 04. December 2018). The data of the objective classification of the prevailing circulation pattern over Europe

Model   is   publicly   available   from   the   German   weather   service   at https://www.dwd.de/DE/leistungen/wetterlagenklassifikation/ wetterlagenklassifikation.html (last access: 04.December 2018). The raw reflectivity volume radar data from the single pol C-Band Doppler radar located in Neuheilenbach is available upon request from the German Weather Service (DWD). The code used to conduct all analyses in this paper is publicly available at https://github.com/KIT-HYD/Radar-based-QPE-using-information-measures.

*Author contribution* UE and MN directly contributed to the conception and design of the work, collected and processed the various data sets and conducted a thorough data quality control, did the data analysis and interpretation and wrote the manuscript.

*Acknowledgements*. We acknowledge support by Deutsche Forschungsgemeinschaft DFG and Open Access Publishing Fund of Karlsruhe Institute of Technology (KIT). This research contributes to the "Catchments As Organized Systems" (CAOS) research group funded by the Deutsche Forschungsgemeinschaft DFG.

We gratefully acknowledge the fruitful cooperation and joint operation of the meteorological observation network within CAOS with Hans-Stefan Bauer and Volker Wulfmeyer.

We gratefully acknowledge the providers of meteorological data used in this study: The German Weather Service DWD for radar and classified circulation pattern data, the Administration des services techniques de l'agriculture ASTA for rain gauge and climate data, and the Hydrometry Service Luxembourg for rain gauge data.

Further on, we especially thank Jean-François Iffly from the Luxembourg Institute of Science and Technology (LIST) for his extensive commitment finding the right locations to put up the instruments and his outstanding help in maintaining them. We

further thank Kirsten Elger from GFZ Helmholtz Centre Potsdam for her patient help with the data publication at the GFZ repository, our colleagues Ralf Loritz, Sibylle Haßler, Conrad Jackisch and Erwin Zehe for fruitful discussions on data analysis and Jan Handwerker from the KIT Institute of Meteorology and Climate Research for help concerning radar meteorology issues. Last but not least we thank the rest of the CAOS team for the warm-hearted project meetings and discussions.



*Competing interests:* The authors declare that they have no conflict of interest.

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
