# Peer review of "Quantitative precipitation estimation with weather radar using a data- and information-based approach"

_Hydrology and Earth System Sciences, 2018_

## Referee Comment (RC1) · Anonymous Referee #1 · 13 Mar 2019

The paper, in general, is quite well written and well structured, and for sure will be interesting to the readers of HESS. Measures of model quality based on information theory add an additional facet to the problem of model assessment and model selection (which traditionally are based mainly on estimates of mean (squared) model error assuming deterministic models). The conducted experiments are comprehensive, very well designed and presented. Illustrative material is adequate. Conclusions are based on evidence from experiments. It can be considered for publication, however there are some issues outlined below which require attention before publication can be recommended. —-

[Figure]

On the references to earlier literature. Immediately after the authors pose the objectives of this work they write: "Comparable approaches have been suggested by Sharma and Mehrotra (2014) and Thiesen et al. (2018)", but in the rest of the paper you do not cite these authors again, and don't mention what are the differences of your approach and the one taken in the papers mentioned. It is suggested to present briefly the essence of the approaches already published, and to explain the advances made in this paper, and formulate its novelty w.r.t. earlier work.

The same can be said about a citation of Yang et al. 2017 or Kirstetter et al. 2015 which is said to be "a similar approach". (It also remains unclear, what this this approach is similar to. Have these researchers also use probabilistic Z-R relationship instead of the deterministic one?)

A general comment: it is suggested, if any reference to earlier work is given, to specify what was done in that work, its main conclusion, and what does this mean for this work.

P 2, L 27: the aim of the paper is formulated: "aim of this paper to suggest and apply a framework which comprises expressing relations among data directly by empirical discrete probability distributions (dpd's), and measuring the strength of relations and remaining uncertainties with measures from Information Theory." However the paper title says that you want to estimate precipitation (QPE). In my view these are two different objective (albeit related). It is also unclear what is "estimation of precipitation" exactly - is it deterministic, or probabilistic? It is therefore suggested to formulate the objectives clearer, and to relate them to the title, and to the section 1.1. "design of experiments".

In Sec 1 there is a block of text, before 1.1, with a number. Suggest to give it a title and number it as well.

P4, L11-13: Eq 1 uses log2 but in the text you mention other options. Please coordinate better. L18: definition of information was about an "event", and on L18 you switch to "signal" (indeed this latter is what the most work on information theory traditionally use).

[Figure]

I suggest to think of a more consistent terminology for this paper, or explain how event and signal relate to each other.

P5 L5: terms expected information and expected uncertainty — here I would use quotes around the terms. L10: why Entropy starts with is the capital letter, and information not? Suggest to be consistent. L15: While –> while

L27: please introduce what is set Y. Are you guessing X, or you are guessing a realisation of X, i.e. xi?

L32: you write Y=yi, but in Eq 3 it us just y

Overall comment on section 2.1: this is a brief introduction to information and entropy, and it uses terminology common for I.T. textbooks. It will be clear to those who know it, but I am not sure all is clear to those who have not used information theory before. It is also somewhat different form the terminology used in the rest of the paper. What is "event"? What is a set X={x1...xn}? Is X a sample from the (same?) distribution? Is its pdf known? Is X a time series as well? An example from hydrometeorology would have helped a lot - assuming, if I understand correctly, that X is some random variable related to rainfall (is its distribution assumed to be known?), and xi are its realisations (is this right?).

P6 L26: "true distribution and a model thereof" - it is first time these terms are used, and a problem of building a model of the true distribution is mentioned. If indeed it is the problem to be solved in this paper, I would suggest to present it earlier.

P7 L3: 2.2. is Methods, but 2.1 was also methods, right? Of Information theory does not belong to Methods? Subsection is entitled: "Data-based models and predictions, information-based model evaluation", so it is also about measures (since "evaluation" is based on measures) - but 2.1 had also "measures for information theory". Suggestion: to coordinate the titles of various sections to prevent overlaps.

Reading this subsection, I was expecting to find the "data-based models", but could

not... It is suggested to clarify what is meant by these and to present them, or not to use this term. (When seeing the term "data-based, or "data-driven" models, I would expect to see a model build on data, using statistical or machine learning techniques and able to make a prediction of a variable (predictant) based on several other variables (predictors).)

L17: "we lose the information about the absolute and relative position of the data tuples in the data set" - unclear. (Do you mean we lose the time stamp of each data tuple (since this are time series)? If so, this is of course always true when a time series is represented as a pdf.)

P8 L1: "statement about the target value" - I don't see a statement about a "value" (value meaning a real valued estimate of the target), just pdf. What is presented on (a very useful and informative) Fig 1 is estimation of predictive uncertainty conditional on the model output (being the range [-2, -2.3]. This uncertainty is estimated based on ALL data across the whole time domain. (Please see e.g. papers by Todini on predictive uncertainty, and I would suggest to add at least one of them.) This is not a model that predicts the target value *for a given time moment*. (But such model what would nice to have of course.)

P17 L25: "to build better models by simply adding more predictors, which according to the Information Inequality (Eq. 4) never hurts." – well, in theory... If we assume that we are building a predictive model (e.g. predicting R on the basis of radar data), in practice more predictors typically means more complex models (more parameters to calibrate/train), and this could be a problem due to the following. (1) You may break the balance between the amount of available data and number of parameters to train, and (2) more model parameters means increasing the dimension of the search space, and it could mean that during training there is a higher chance to be stuck in a local minimum (e.g. if MLP neural networks are used). So there are good practical reasons to avoid having too many predictors, and (3) more complex model may overfit and not be accurate on cross-validation sets.

P18 L1: I am not sure I understand how overfitting relates to curse of dimensionality by Bellman.

— To summarize the most important comments (and there are more above):

1. The title promises "Quantitative precipitation estimation using a data-based approach". Perhaps I missed something, but I have not found a data-based model that makes a quantitative estimation of rainfall using radar data. Yes, analysis of distributions of radar data help to provide informative uncertainty estimates of precipitation (and assess the overall information gain) but in my view it is not what is typically understood by QPE (deterministic, and possibly with uncertainty estimates on top).

2. Suggested to present clearly innovation w.r.t. to earlier work.

3. Introduction: suggested to formulate the objectives clearer, and to relate them to the six experiments.

4. On the ways to measure the quality of predictive models. What does real practice need? In essence, the probability distribution-based measures (including the ones using the information theory) give an estimate how far are distributions from each other, rather than measuring the distance between time series, i.e. how far are the deterministic model predictions from (ground truth) observations (on average, or for particular important time moments). In my view it is important to stress this difference. So decision on what measures to use depends on the needs of practitioners and the corresponding goal setting. I would suggest that the best is use both since they measure different things.

5. As far as I understand, the authors used All data to estimate the information gain (and hence to "build" a model allowing for QPE). In other words, there is no data left for cross validation and for testing. If so, this does not correspond to the theory of data-based model building. Is this right?

Editorial comments:

Minor editing of English may be desirable, e.g.: Besides rain gauges with its own limitations: its –> their

Radar data have among other been used for urban hydrology –> Among other data sources, radar data have been used in urban hydrology

its use relies on some sometimes more, sometimes less justified assumptions –> its use relies on some assumptions, which are sometimes justified and sometimes not,

Much work has since then been done –> Much work has since been done

framework which comprises expressing relations among data directly by –> framework which would use relationships between data expressed as the

I suggest shorten some long sentences, and to split long paragraphs (e.g. the first one in the Introduction).

—- The authors are kindly invited to address these comments, revise the paper (or provide a rebuttal), and to resubmit it.

---

## Short Comment (SC1) · 25 Mar 2019

**Responses to comments posted by Referee #1**

We thank the first referee for reviewing our article and providing his fruitful feedbacks. They identify some unclear issues and help to improve the presentation of our research. In the following, we answer to all of the comments one by one. The Referee comments are in blue.

Comment 1: The paper, in general, is quite well written and well structured, and for sure will be interesting to the readers of HESS. Measures of model quality based on information theory add an additional facet to the problem of model assessment and model selection (which traditionally are based mainly on estimates of mean (squared) model error assuming deterministic models). The conducted experiments are comprehensive, very well designed and presented. Illustrative material is adequate. Conclusions are based on evidence from experiments. It can be considered for publication, however there are some issues outlined below which require attention before publication can be recommended.

Reply 1: We thank the referee for his generally positive evaluation of our manuscript.

Comment 2: On the references to earlier literature. Immediately after the authors pose the objectives of this work they write: "Comparable approaches have been suggested by Sharma and Mehrotra (2014) and Thiesen et al. (2018)", but in the rest of the paper you do not cite these authors again, and don't mention what are the differences of your approach and the one taken in the papers mentioned. It is suggested to present briefly the essence of the approaches already published, and to explain the advances made in this paper, and formulate its novelty w.r.t. earlier work.

Reply 2: Agreed. We will do so in a revised version of the manuscript.

Comment 3: The same can be said about a citation of Yang et al. 2017 or Kirstetter et al. 2015 which is said to be "a similar approach". (It also remains unclear, what this this approach is similar to. Have these researchers also use probabilistic Z-R relationship instead of the deterministic one?)

Reply 3: Agreed. We will do so in a revised version of the manuscript.

Comment 4: A general comment: it is suggested, if any reference to earlier work is given, to specify what was done in that work, its main conclusion, and what does this mean for this work.

Reply 4: Agreed. See Replies 2 and 3. We will do so in a revised version of the manuscript. We will briefly describe the results and applied methods of the works cited. In addition we will more explicitly show how our manuscript compares and contrasts to the previously published and cited work.

Comment 5:  P 2, L 27: the aim of the paper is formulated: "aim of this paper to suggest and apply a framework which comprises expressing relations among data directly by empirical discrete probability distributions (dpd's), and measuring the strength of relations and remaining uncertainties with measures from Information Theory." However the paper title says that you want to estimate precipitation (QPE). In my view these are two different objective (albeit related). It is also unclear what is "estimation of precipitation" exactly - is it deterministic, or probabilistic? It is therefore suggested to formulate the objectives clearer, and to relate them to the title, and to the section 1.1. "design of experiments".

Reply 5: Agreed. In a revised version of the manuscript, we will explain more clearly that with the paper we follow indeed two, albeit related objectives: Present the information-based and probabilistic approach to estimation/prediction in general, and show how it works for QPE in particular. To further comment on this: With our approach we do not want to provide a deterministic, single-valued rainrate, but promote the use of probabilistic QPE, which adequately reflects the (considerable) intrinsic uncertainties related to radar-based QPE. The history of radarmeteorology shows that the most important part is the quality control and the error handling. With our approach we want to provide a probable rain rate value range and distribution, which for users has the added value of knowing intrinsic uncertatinties compared to a single-valued QPE value. This is similar to ensemble approaches in operational weather forecasting. By knowing a predictive distribution rather than a single value the user will make better-informed decisions, especially in the case of extreme events. And if the user requires a single-valued statement: a distribution can always be collapsed to a single value, e.g. by calculating the mean or mode, while this is not possible in the opposite direction.

Comment 6:  In Sec 1 there is a block of text, before 1.1, with a number. Suggest to give it a title and number it as well.

Reply 6: Agreed. We will do so in a revised version of the manuscript.

Comment 7: P4, L11-13: Eq 1 uses log2 but in the text you mention other options. Please coordinate better. L18: definition of information was about an "event", and on L18 you switch to "signal" (indeed this latter is what the most work on information theory traditionally use). I suggest to think of a more consistent terminology for this paper, or explain how event and signal relate to each other.

Reply 7: Agreed. We will move the first sentence after Eq 1 up, to make clear that while several units for information exist, we will stick to [bit]. Also, in a revised version of the manuscript we will introduce both terms 'event' and 'signal' once to explain that both refer to 'the outcome of a random experiment, i.e. drawing randomly from a known distribution', but then will stick to 'signal' throughout the text.

Comment 8: P5 L5: terms expected information and expected uncertainty — here I would use quotes around the terms. L10: why Entropy starts with is the capital letter, and information not? Suggest to be consistent. L15: While –> while

Reply 8: P5L5: Agreed, will do. L10: Agreed, we will use both 'Entropy' and 'Information' with capital letters throughout the text. L15: Agreed, will do.

Comment 9: L27: please introduce what is set Y. Are you guessing X, or you are guessing a realisation of X, i.e. xi?

Reply 9: As explained in the sentence starting in L28, Y refers to available (a priori known) data used to guess an unknown target quantity of interest, X. To do so, both the general relation between X and Y must be known from a joint learning data set {X,Y} with paired data $x_1,y_1$ $x_2,y_2$ etc. ; and a particular observation of Y (i.e. y) to guess the particular, related realisation of X (i.e. x). We agree that subscript 'i' here may lead to confusion and will distinguish in a revised version of the manuscript the set from a particular value coming from that set by capital and small letters only, e.g. X, x. Also, we will illustrate these things in more detail with an example using hydrometeorological data (i.e. we will elaborate the sentence starting in L28).

Comment 10: L32: you write Y=yi, but in Eq 3 it us just y

Reply 10: Please also see our reply to Comment 9: We agree that subscript 'i' here may lead to confusion and will distinguish in a revised version of the manuscript the set from a particular value coming from that set by capital and small letters only, e.g. X, x.

Comment 11: Overall comment on section 2.1: this is a brief introduction to information and entropy, and it uses terminology common for I.T. textbooks. It will be clear to those who know it, but I am not sure all is clear to those who have not used information theory before. It is also somewhat different form the terminology used in the rest of the paper. What is "event"? What is a set X={x1...xn}? Is X a sample from the (same?) distribution? Is its pdf known? Is X a time series as well? An example from hydrometeorology would have helped a lot - assuming, if I understand correctly, that X is some random variable related to rainfall (is its distribution assumed to be known?), and xi are its realisations (is this right?).

Reply 11: The referee's interpretation of X, $x_i$ and the general procedure is correct. We agree that we need to better explain these things in the text and will do so according to our suggestions in Reply's 9 and 10.

Comment 12: P6 L26: "true distribution and a model thereof" - it is first time these terms are used, and a problem of building a model of the true distribution is mentioned. If indeed it is the problem to be solved in this paper, I would suggest to present it earlier.

Reply 12: Good point. To express more clearly what we mean by 'true distribution' and 'a model thereof', we will introduce these terms in the previous paragraph about Cross Entropy (after the sentence starting in L13) and add an illustrative example.

Comment 13: P7 L3: 2.2. is Methods, but 2.1 was also methods, right? Of Information theory does not belong to Methods? Subsection is entitled: "Data-based models and predictions, information-based model evaluation", so it is also about measures (since "evaluation" is based on measures) - but 2.1 had also "measures for information theory". Suggestion: to coordinate the titles of various sections to prevent overlaps.

Reply 13: Good point. We will retitle the subsection, to prevent apparent overlaps. Our new title for subsection 2.2 (P7 L3) will be "Modelling and evaluation strategy").

Comment 14: Reading this subsection, I was expecting to find the "data-based models", but could not... It is suggested to clarify what is meant by these and to present them, or not to use this term. (When seeing the term "data-based, or "data-driven" models, I would expect to see a model build on data, using statistical or machine learning techniques and able to make a prediction of a variable (predictant) based on several other variables (predictors).)

Reply 14: Thanks for the comment, which indicates we need to explain better what we mean by 'data-based model' in this paper: The model is simply the empirical, multivariate, discrete frequency (or probability when normalized) distribution derived from all available data. Its input is values for all predictors, and it returns a conditional (on the values of the predictors) distribution of the predictant. We will explain this better in section 2.2 of a revised version of the manuscript.

Comment 15: L17: "we lose the information about the absolute and relative position of the data tuples in the data set" - unclear. (Do you mean we lose the time stamp of each data tuple (since this are time series)? If so, this is of course always true when a time series is represented as a pdf.)

Reply 15: Yes, it means we lose the information contained in the time stamp (i.e. absolute position and relative order of the data), and this not specific to our approach but always true if a time series is mapped to a pdf. We will explain this in more detail in a revised version of the manuscript.

Comment 16: P8 L1: "statement about the target value" - I don't see a statement about a "value" (value meaning a real valued estimate of the target), just pdf. What is presented on (a very useful and informative) Fig 1 is estimation of predictive uncertainty conditional on the model output (being the range [-2, -2.3]. This uncertainty is estimated based on ALL data across the whole time domain. (Please see e.g. papers by Todini on predictive uncertainty, and I would

suggest to add at least one of them.) This is not a model that predicts the target value *for a given time moment*. (But such model what would nice to have of course.)

Reply 16: Indeed the 'value' statement in our case does not contain a single value, but a pdf, which to us is a joint statement about the predicted target value AND at the same time its predictive uncertainty. If for any reason the receiver of such a prediction cannot use the pdf directly, then its two components (value and uncertainty) can easily be extracted by e.g. calculating its mean (or mode) and the variance (or entropy). Please see also our related Reply 5. The referee is right in stating the uncertainty is based on ALL data across the whole time domain, or, more specific, based on all data with indistinguishable (in the light of the chosen binning) predictor values (in Fig. 1 predictor values in range [-2,-2.3]). So from a predictor value point of view, these predictive situations are all the same, and should hence also lead to the same values of the predictant. In reality they don't, because the predictors don not represent all factors influencing the predictant's value, so what we get is a predictive distribution rather than a single value. Thank you for pointing us at the related work of Ezio Todini, we will do a literature review and will cite related literature in a revised version of the manuscript.

Comment 17: P17 L25: "to build better models by simply adding more predictors, which according to the Information Inequality (Eq. 4) never hurts." – well, in theory... If we assume that we are building a predictive model (e.g. predicting R on the basis of radar data), in practice more predictors typically means more complex models (more parameters to calibrate/train), and this could be a problem due to the following. (1) You may break the balance between the amount of available data and number of parameters to train, and (2) more model parameters means increasing the dimension of the search space, and it could mean that during training there is a higher chance to be stuck in a local minimum (e.g. if MLP neural networks are used). So there are good practical reasons to avoid having too many predictors, and (3) more complex model may overfit and not be accurate on cross-validation sets.

Reply 17: We completely agree with the referee's reasons against using more and more predictors and that overfitting should be avoided. This holds for parametric models (models with a predefined structure plus parameters estimated with a training data set) as described by the referee, but also for the 'conditional pdf' approach we use. While in our case there are no parameters to train, the choices we need to make are the type and number of predictors and the related binning. In Figs 3 and 4, we show the relation between number of predictors and required data to adequately 'train' (in our case it's rather 'construct') the model, which is essentially a great number of split sample tests for different sizes of the learning data set. We suggest including in a revised version of the manuscript a more extended discussion about the potential problems of adding more and more predictors along the lines of the referee comment and our reply.

Comment 18: P18 L1: I am not sure I understand how overfitting relates to curse of dimensionality by Bellman.

Reply 18: We agree, this is misleading. For brevity and clarity, we suggest to restrict the discussion to 'overfitting' here.

Comment 19: To summarize the most important comments (and there are more above):

1.  The title promises "Quantitative precipitation estimation using a data-based approach". Perhaps I missed something, but I have not found a data-based model that makes a quantitative estimation of rainfall using radar data. Yes, analysis of distributions of radar data help to provide informative uncertainty estimates of precipitation (and assess the overall information gain) but in my view it is not what is typically understood by QPE (deterministic, and possibly with uncertainty estimates on top).
2.  Suggested to present clearly innovation w.r.t. to earlier work.
3.  Introduction: suggested to formulate the objectives clearer, and to relate them to the six experiments.
4.  On the ways to measure the quality of predictive models. What does real practice need? In essence, the probability distribution-based measures (including the ones using the information theory) give an estimate how far are distributions from each other, rather than measuring the distance between time series, i.e. how far are the deterministic model predictions from (ground truth) observations (on average, or for particular important time moments). In my view it is important to stress this difference. So decision on what measures to use depends on the needs of practitioners and the corresponding goal setting. I would suggest that the best is use both since they measure different things.
5.  5. As far as I understand, the authors used All data to estimate the information gain (and hence to "build" a model allowing for QPE). In other words, there is no data left for cross validation and for testing. If so, this does not correspond to the theory of data-based model building. Is this right?

Reply 19

1.  We hope we could suitably answer this point in replies 5 and 16.
2.  We hope we could suitably answer this point in replies 2, 3 and 4
3.  We hope we could suitably answer this point in reply 5.
4.  We agree that quality measures should be selected with the intended use of the model in mind, and established quality measures expressing distances in the units of the data (e.g. Nash-Sutcliffe efficiency, bias, RMSE etc.) are clearly valuable. Problems arise in the case of multivariate data and models: How to combine distance measures expressed in different units? Possible solutions involve normalization of each distance measure to [0,1] scales or choice of a weighted combination scheme. Using quality measures in the probability domain

however, such as the information measures we apply, has the advantage of being independent of the units of the data, which facilitates comparison and combination. Also, both Cross Entropy and Kullback-Leibler divergence as presented in the text can be used as distance measures between a single-valued observation/ground truth and a probabilistic prediction. In this particular case, one of the two distributions (the observation) is simply a distribution with p=1 for a single bin, and p=0 for all others. Kullback-Leibler divergence then becomes a measure of likelihood of the observation given the model.

5.  We indeed used all data to construct the 'perfect models'. However, in order to evaluate the effect of overfitting, we used a very large number of modified split-sampling tests as explained in section 2.2 (sampling strategy), 3.2 and displayed in Figs 4 and 5. An example: Fig 4, green line, sample size 2000: The cross entropy is 1.75. This value was obtained as follows: From all available data, 2000 value pairs of RR0 and dBZ1500Rad were randomly chosen and used to construct the multivariate, discrete pdf's which manifest our model. This model was then applied to ALL data (including the 2000 for training) and Cross Entropy was calculated to the 'perfect model' based on all data. The entire procedure was repeated 500 times. The value in the Figure shows the mean of all 500 Cross Entropies. This is not a classical split sample test where calibration and validation data set are completely separate. However, the advantage we see in our approach is that we can express total uncertainty as an additive combination of two effects: Uncertainty due to the limited information content of the predictor about the predictant (which is Cross Entropy using all data, i.e. Cross Entropy at the right end of the plot), and uncertainty because we use a model constructed from only a limited data set. We suggest that while this is not a standard way of model evaluation, it offers useful ways to look at magnitude and sources of uncertainty. In this context, please also see our replies 16 and 17.

Editorial comments

Comment 20: Minor editing of English may be desirable, e.g.:

- Besides rain gauges with its own limitations: its –> their
  Reply: Agreed. We will change this in a revised version of the manuscript
- Radar data have among other been used for urban hydrology –> Among other data sources, radar data have been used in urban hydrology
  Reply: Agreed. We will change this in a revised version of the manuscript
- its use relies on some sometimes more, sometimes less justified assumptions –> its use relies on some assumptions, which are sometimes justified and sometimes not,
  Reply: Agreed. We will change this in a revised version of the manuscript
- Much work has since then been done –> Much work has since been done
  Reply: Agreed. We will change this in a revised version of the manuscript

- framework which comprises expressing relations among data directly by –> framework which would use relationships between data expressed as the
  Reply: Agreed. We will change this in a revised version of the manuscript
- I suggest shorten some long sentences, and to split long paragraphs (e.g. the first one in the Introduction).
  Reply: Agreed. We will change this in a revised version of the manuscript

We thank the referee for the editing suggestions. Besides this, there will be a final copy editing by HESS before the paper is published.

Malte Neuper and Uwe Ehret

---

## Referee Comment (RC2) · Anonymous Referee #2 · 25 Apr 2019

General comment:

This article is very interesting and I think it could be published with some minor corrections. The approach used is complementary to what is commonly done. It makes it possible to identify interesting predictors to be taken into account in order to improve the quantitative estimation of rainfall. However, even if it can be a guide, I have some doubts about the ability of Information Theory to go further, that is make it possible, in practice, to improve quantitative estimation. The article is well written. A reader who not used with information theory is able, with a few "round trips" in the article, to understand the main ideas.

Remarks on the substance:

* Page 2, line 24: It seems to me that the work of Vignal et al (1999) and (2000) only addresses the variability of Z from high altitude to the ground but not directly the variability of the Z-R relationship along the vertical.

* Page 12, line 11: the reflectivity factors of the MRR, due to the short wavelength used, can also sometimes be greatly attenuated, with consequences on the retrieved DSD.

* Page 12, line 11: why is the 3.5 moment used and not (the most commonly used) 3.67? Could you add a reference?

* Page 17, Line 11: A 15.3% reduction in uncertainty when RR0 is conditioned by dBZ1500Rad seems quite low, doesn't it? This proves that radar reflectivity is one of the most important sources of information but that there is a high variability in the Z-R relationship that limits entropy reduction. This explanation is suggested later (experiment 3) but could perhaps already be mentioned here. I think we are also suffering here from the effects of PVR and perhaps even attenuation for high intensities.

* Page 24, Line 3-5: I understand that DKL=3.43 bits and 5.04 correspond to the application of the Z-R Marshall Palmer relationship and that DKL=2.69 and 4.30 correspond to the optimized Z-R relationship. But I don't understand what differentiates 3.43 from 2.69 (and 5.04 from 4.30): what are the differences in terms of condition applied?

* Page 26, Line 22: after "attributed to the ambiguous relationship between radar reflectivity and rain rate", I think that "due to the natural variability of Drop Size Distribution" may be added.

* Section 3.6: Why kriging (often used method) was not considered for the rain-gauge approach?

Remarks on the form of the document:

* The publication of Cecinate et al (2017) cited on page 2 do not appear in the list of references.

* Section 1.1, page 3, is a little "orphaned". It may not be necessary to distinguish this paragraph from the introduction. Its content could appear just before the announcement of the plan (between lines 10 and 11 on page 3).

* Table 1: I propose to replace "[min, max]" by"[center of the min bin, center of max bin]" or something like that.

* Figures 4, 5, 6 and 7: I think that the texts of the legends should be enlarged.

* Figure 6: Wouldn't the figure in a log-log frame be more readable and give less weight to very weak reflectivities?

* Table 3: I propose to replace "RR0" by "RR0Dis" to highlight that disrometer is used (as for dBZ0Dis).
* * *

---

## Author Response (AR1)

**Responses to comments posted by Referee #1**

We thank the first referee for reviewing our article and providing his fruitful feedbacks. They identify some unclear issues and help to improve the presentation of our research. In the following, we answer to all of the comments one by one. The Referee comments are in blue.

The location of the specific author's changes in the marked-up manuscript version (track changes) are given in green.

Comment 1: The paper, in general, is quite well written and well structured, and for sure will be interesting to the readers of HESS. Measures of model quality based on information theory add an additional facet to the problem of model assessment and model selection (which traditionally are based mainly on estimates of mean (squared) model error assuming deterministic models). The conducted experiments are comprehensive, very well designed and presented. Illustrative material is adequate. Conclusions are based on evidence from experiments. It can be considered for publication, however there are some issues outlined below which require attention before publication can be recommended.

Reply 1: We thank the referee for his generally positive evaluation of our manuscript.

Comment 2: On the references to earlier literature. Immediately after the authors pose the objectives of this work they write: "Comparable approaches have been suggested by Sharma and Mehrotra (2014) and Thiesen et al. (2018)", but in the rest of the paper you do not cite these authors again, and don't mention what are

the differences of your approach and the one taken in the papers mentioned. It is suggested to present briefly the essence of the approaches already published, and to explain the advances made in this paper, and formulate its novelty w.r.t. earlier work.

Reply 2: Agreed. We will do so in a revised version of the manuscript.

Author's change: P 3, L 4

Comment 3: The same can be said about a citation of Yang et al. 2017 or Kirstetter et al. 2015 which is said to be "a similar approach". (It also remains unclear, what this this approach is similar to. Have these researchers also use probabilistic Z-R relationship instead of the deterministic one?)

Reply 3: Agreed. We will do so in a revised version of the manuscript.

Author's change: P 3, L 9

Comment 4: A general comment: it is suggested, if any reference to earlier work is given, to specify what was done in that work, its main conclusion, and what does this mean for this work.

Reply 4: Agreed. See Replies 2 and 3. We will do so in a revised version of the manuscript. We will briefly describe the results and applied methods of the works cited. In addition we will more explicitly show how our manuscript compares and contrasts to the previously published and cited work.

Comment 5: P 2, L 27: the aim of the paper is formulated: "aim of this paper to suggest and apply a framework which comprises expressing relations among data directly by empirical discrete probability distributions (dpd's), and measuring the strength of relations and remaining uncertainties with measures from Information Theory." However the paper title says that you want to estimate precipitation (QPE). In my view these are two different objective (albeit related). It is also unclear what is "estimation of precipitation" exactly - is it deterministic, or probabilistic? It is therefore suggested to formulate the objectives clearer, and to relate them to the title, and to the section 1.1. "design of experiments".

Reply 5: Agreed. In a revised version of the manuscript, we will explain more clearly that with the paper we follow indeed two, albeit related objectives: Present the information-based and probabilistic approach to estimation/prediction in general, and show how it works for QPE in particular. To further comment on this: With our approach we do not want to provide a deterministic, single-valued rainrate, but promote the use of probabilistic QPE, which adequately reflects the (considerable) intrinsic uncertainties related to radar-based QPE. The history of radarmeteorology shows that the most important part is the quality control and the error handling. With our approach we want to provide a probable rain rate value range and distribution, which for users has the added value of knowing intrinsic uncertatinties compared to a single-valued QPE value. This is similar to ensemble approaches in operational weather forecasting. By knowing a predictive distribution rather than a single value the user will make

better-informed decisions, especially in the case of extreme events. And if the user requires a single-valued statement: a distribution can always be collapsed to a single value, e.g. by calculating the mean or mode, while this is not possible in the opposite direction.

Author's change: P 3, L 16

Comment 6: In Sec 1 there is a block of text, before 1.1, with a number. Suggest to give it a title and number it as well.

Reply 6: Agreed. We will do so in a revised version of the manuscript.

Author's change: P 2, L 6

Comment 7: P4, L11-13: Eq 1 uses log2 but in the text you mention other options. Please coordinate better. L18: definition of information was about an "event", and on L18 you switch to "signal" (indeed this latter is what the most work on information theory traditionally use). I suggest to think of a more consistent terminology for this paper, or explain how event and signal relate to each other.

Reply 7: Agreed. We will move the first sentence after Eq 1 up, to make clear that while several units for information exist, we will stick to [bit]. Also, in a revised version of the manuscript we will introduce both terms 'event' and 'signal' once to explain that both refer to 'the outcome of a random experiment, i.e. drawing randomly from a known distribution', but then will stick to 'signal' throughout the text.

Author's change: P 4, L 31 – P 5, L 4

Comment 8: P5 L5: terms expected information and expected uncertainty — here I would use quotes around the terms. L10: why Entropy starts with is the capital letter, and information not? Suggest to be consistent. L15: While –> while

Reply 8: P5L5: Agreed, will do. L10: Agreed, we will use both 'Entropy' and 'Information' with capital letters throughout the text. L15: Agreed, will do.

Author's change: Quotes: P 5 L 27 - 'Information' with capital letter changed throughout the text (e.g. P 5, L 11, 12, 13, 14, 17, 19, 27, 31) – While -> while (P6 L 6)

Comment 9: L27: please introduce what is set Y. Are you guessing X, or you are guessing a realisation of X, i.e. xi?

Reply 9: As explained in the sentence starting in L28, Y refers to available (a priori known) data used to guess an unknown target quantity of interest, X. To do so, both the general relation between X and Y must be known from a joint learning data set {X,Y} with paired data x1,y1 x2,y2 etc. ; and a particular observation of Y (i.e. y) to guess the particular, related realisation of X (i.e. x). We agree that subscript 'i' here may lead to confusion and will distinguish in a revised version of the manuscript the set from a particular value coming from that set by capital and small letters only, e.g. X, x. Also, we will illustrate these things in more detail with an example using hydrometeorological data (i.e. we will elaborate the sentence starting in L28).

Author's change: P 6, L 18 – P 7, L 2

Comment 10: L32: you write Y=yi, but in Eq 3 it us just y

Reply 10: Please also see our reply to Comment 9: We agree that subscript 'i' here may lead to confusion and will distinguish in a revised version of the manuscript the set from a particular value coming from that set by capital and small letters only, e.g. X, x.

Author's change: P 6, L 24 – P 7, L 4

Comment 11: Overall comment on section 2.1: this is a brief introduction to information and entropy, and it uses terminology common for I.T. textbooks. It will be clear to those who know it, but I am not sure all is clear to those who have not used information theory before. It is also somewhat different form the terminology used in the rest of the paper. What is "event"? What is a set X={x1...xn}? Is X a sample from the (same?) distribution? Is its pdf known? Is X a time series as well? An example from hydrometeorology would have helped a lot - assuming, if I understand correctly, that X is some random variable related to rainfall (is its distribution assumed to be known?), and xi are its realisations (is this right?).

Reply 11: The referee's interpretation of X, xi and the general procedure is correct. We agree that we need to better explain these things in the text and will do so according to our suggestions in Reply's 9 and 10.

Comment 12: P6 L26: "true distribution and a model thereof" - it is first time these terms are used, and a problem of building a model of the true distribution is mentioned. If indeed it is the problem to be solved in this paper, I would suggest to present it earlier.

Reply 12: Good point. To express more clearly what we mean by 'true distribution' and 'a model thereof', we will introduce these terms in the previous paragraph about Cross Entropy (after the sentence starting in L13) and add an illustrative example.

Author's change: P 7, L 13 - 29

Comment 13: P7 L3: 2.2. is Methods, but 2.1 was also methods, right? Of Information theory does not belong to Methods? Subsection is entitled: "Data-based models and predictions, information-based model evaluation", so it is also about measures (since "evaluation" is based on measures) - but 2.1 had also "measures for information theory". Suggestion: to coordinate the titles of various sections to prevent overlaps.

Reply 13: Good point. We will retitle the subsection, to prevent apparent overlaps. Our new title for subsection 2.2 (P7 L3) will be "Modelling and evaluation strategy").

Author's change: P 8, L 5

Comment 14: Reading this subsection, I was expecting to find the "data-based models", but could not... It is suggested to clarify what is meant by these and to present them, or not to use this term. (When seeing the term "data-based, or "data-driven" models, I would expect to see a model build on data, using statistical or machine learning techniques and able to make a prediction of a variable (predictant) based on several other variables (predictors).)

Reply 14: Thanks for the comment, which indicates we need to explain better what we mean by 'data-based model' in this paper: The model is simply the empirical, multivariate, discrete frequency (or probability when normalized) distribution derived from all available data. Its input is values for all predictors, and it returns a conditional (on the values of the predictors) distribution of the predictant. We will explain this better in section 2.2 of a revised version of the manuscript.

Author's change: P 9, L 3 -17

Comment 15: L17: "we lose the information about the absolute and relative position of the data tuples in the data set" - unclear. (Do you mean we lose the time stamp of each data tuple (since this are time series)? If so, this is of course always true when a time series is represented as a pdf.)

Reply 15: Yes, it means we lose the information contained in the time stamp (i.e. absolute position and relative order of the data), and this not specific to our

approach but always true if a time series is mapped to a pdf. We will explain this in more detail in a revised version of the manuscript.

Comment 16: P8 L1: "statement about the target value" - I don't see a statement about a "value" (value meaning a real valued estimate of the target), just pdf. What is presented on (a very useful and informative) Fig 1 is estimation of predictive uncertainty conditional on the model output (being the range [-2, -2.3]. This uncertainty is estimated based on ALL data across the whole time domain. (Please see e.g. papers by Todini on predictive uncertainty, and I would suggest to add at least one of them.) This is not a model that predicts the target value *for a given time moment*. (But such model what would nice to have of course.)

Reply 16: Indeed the 'value' statement in our case does not contain a single value, but a pdf, which to us is a joint statement about the predicted target value AND at the same time its predictive uncertainty. If for any reason the receiver of such a prediction cannot use the pdf directly, then its two components (value and uncertainty) can easily be extracted by e.g. calculating its mean (or mode) and the variance (or entropy). Please see also our related Reply 5. The referee is right in stating the uncertainty is based on ALL data across the whole time domain, or, more specific, based on all data with indistinguishable (in the light of the chosen binning) predictor values (in Fig. 1 predictor values in range [-2,-2.3]). So from a predictor value point of view, these predictive situations are all the same, and should hence also lead to the same values of the predictant. In reality they don't, because the predictors don not represent all factors influencing the predictant's

value, so what we get is a predictive distribution rather than a single value. Thank you for pointing us at the related work of Ezio Todini, we will do a literature review and will cite related literature in a revised version of the manuscript.

Author's change: P 9, L 9 - 14

Comment 17: P17 L25: "to build better models by simply adding more predictors, which according to the Information Inequality (Eq. 4) never hurts." – well, in theory... If we assume that we are building a predictive model (e.g. predicting R on the basis of radar data), in practice more predictors typically means more complex models (more parameters to calibrate/train), and this could be a problem due to the following. (1) You may break the balance between the amount of available data and number of parameters to train, and (2) more model parameters means increasing the dimension of the search space, and it could mean that during training there is a higher chance to be stuck in a local minimum (e.g. if MLP neural networks are used). So there are good practical reasons to avoid having too many predictors, and (3) more complex model may overfit and not be accurate on cross-validation sets.

Reply 17: We completely agree with the referee's reasons against using more and more predictors and that overfitting should be avoided. This holds for parametric models (models with a predefined structure plus parameters estimated with a training data set) as described by the referee, but also for the 'conditional pdf' approach we use. While in our case there are no parameters to train, the choices we need to make are the type and number of predictors and the related binning.

In Figs 3 and 4, we show the relation between number of predictors and required data to adequately 'train' (in our case it's rather 'construct') the model, which is essentially a great number of split sample tests for different sizes of the learning data set. We suggest including in a revised version of the manuscript a more extended discussion about the potential problems of adding more and more predictors along the lines of the referee comment and our reply.

Author's change: P 20, L 23 - 25

Comment 18: P18 L1: I am not sure I understand how overfitting relates to curse of dimensionality by Bellman.

Reply 18: We agree, this is misleading. For brevity and clarity, we suggest to restrict the discussion to 'overfitting' here.

Author's change: P 20, L 23

Comment 19: To summarize the most important comments (and there are more above):

1. The title promises "Quantitative precipitation estimation using a data-based approach". Perhaps I missed something, but I have not found a data-based model that makes a quantitative estimation of rainfall using radar data. Yes, analysis of distributions of radar data help to provide informative uncertainty estimates of precipitation (and assess the overall information gain) but in my view it is not what is typically understood by QPE (deterministic, and possibly with uncertainty estimates on top).

2. Suggested to present clearly innovation w.r.t. to earlier work.

3. Introduction: suggested to formulate the objectives clearer, and to relate them to the six experiments.

4. On the ways to measure the quality of predictive models. What does real practice need? In essence, the probability distribution-based measures (including the ones using the information theory) give an estimate how far are distributions from each other, rather than measuring the distance between time series, i.e. how far are the deterministic model predictions from (ground truth) observations (on average, or for particular important time moments). In my view it is important to stress this difference. So decision on what measures to use depends on the needs of practitioners and the corresponding goal setting. I would suggest that the best is use both since they measure different things.

5. As far as I understand, the authors used All data to estimate the information gain (and hence to "build" a model allowing for QPE). In other words, there is no data left for cross validation and for testing. If so, this does not correspond to the theory of data-based model building. Is this right?

Reply 19

1. We hope we could suitably answer this point in replies 5 and 16.

2. We hope we could suitably answer this point in replies 2, 3 and 4

3. We hope we could suitably answer this point in reply 5.

4. We agree that quality measures should be selected with the intended use of the model in mind, and established quality measures expressing distances in the units of the data (e.g. Nash-Sutcliffe efficiency, bias, RMSE etc.) are clearly valuable. Problems arise in the case of multivariate data and models: How to combine

distance measures expressed in different units? Possible solutions involve normalization of each distance measure to [0,1] scales or choice of a weighted combination scheme. Using quality measures in the probability domain however, such as the information measures we apply, has the advantage of being independent of the units of the data, which facilitates comparison and combination. Also, both Cross Entropy and Kullback-Leibler divergence as presented in the text can be used as distance measures between a single-valued observation/ground truth and a probabilistic prediction. In this particular case, one of the two distributions (the observation) is simply a distribution with p=1 for a single bin, and p=0 for all others. Kullback-Leibler divergence then becomes a measure of likelihood of the observation given the model.

5. We indeed used all data to construct the 'perfect models'. However, in order to evaluate the effect of overfitting, we used a very large number of modified split-sampling tests as explained in section 2.2 (sampling strategy), 3.2 and displayed in Figs 4 and 5. An example: Fig 4, green line, sample size 2000: The cross entropy is 1.75. This value was obtained as follows: From all available data, 2000 value pairs of RR0 and dBZ1500Rad were randomly chosen and used to construct the multivariate, discrete pdf's which manifest our model. This model was then applied to ALL data (including the 2000 for training) and Cross Entropy was calculated to the 'perfect model' based on all data. The entire procedure was repeated 500 times. The value in the Figure shows the mean of all 500 Cross Entropies. This is not a classical split sample test where calibration and validation data set are completely separate. However, the advantage we see in our approach is that we can express total uncertainty as an additive combination of two effects: Uncertainty due to the limited information content of the predictor about the predictant (which is Cross Entropy using all data, i.e. Cross Entropy at the right end

of the plot), and uncertainty because we use a model constructed from only a limited data set. We suggest that while this is not a standard way of model evaluation, it offers useful ways to look at magnitude and sources of uncertainty. In this context, please also see our replies 16 and 17.

Editorial comments

Comment 20: Minor editing of English may be desirable, e.g.:

- Besides rain gauges with its own limitations: its –> their

Reply: Agreed. We will change this in a revised version of the manuscript

Author's change: P 2 , L 8

- Radar data have among other been used for urban hydrology –> Among other data sources, radar data have been used in urban hydrology

Reply: Agreed. We will change this in a revised version of the manuscript

Author's change: P 2, L 10

- its use relies on some sometimes more, sometimes less justified assumptions –> its use relies on some assumptions, which are sometimes justified and sometimes not,

Reply: Agreed. We will change this in a revised version of the manuscript

Author's change: P 2, L 17

- Much work has since then been done –> Much work has since been done

Reply: Agreed. We will change this in a revised version of the manuscript

Author's change: P 2, L 26

- framework which comprises expressing relations among data directly by –> framework which would use relationships between data expressed as the

Reply: Agreed. We will change this in a revised version of the manuscript

Author's change: P 3, L 1

- I suggest shorten some long sentences, and to split long paragraphs (e.g. the first one in the Introduction).

Reply: Agreed. We will change this in a revised version of the manuscript

We thank the referee for the editing suggestions. Besides this, there will be a final copy editing by HESS before the paper is published.

Malte Neuper and Uwe Ehret

**Responses to comments posted by Referee #2**

We thank the second referee for reviewing our article and providing his fruitful feedbacks. In the following, we answer to all of the comments one by one. The Referee comments are in blue.

Comment 1: Page 2, line 24: It seems to me that the work of Vignal et al (1999) and (2000) only addresses the variability of Z from high altitude to the ground but not directly the variability of the Z-R relationship along the vertical.

Reply 1: We agree. To make things clearer, we will better explain the content of Vignal et al. (1999) and (2000) and will add another reference which more closely addresses the variability of the Z-R relationship to the ground: Peters, G., Münster, H., Clemens, M., and Wagner, A., (2005): Profiles of raindrop size distributions as retrieved by microrain radars. J. Appl. Meteor., 44, 1930–1949,https://doi.org/10.1175/JAM2316.1.

Author's change: P 2, L 27 - 29

Comment 2: Page 12, line 11: the reflectivity factors of the MRR, due to the short wavelength used, can also sometimes be greatly attenuated, with consequences on the retrieved DSD.

Reply 2: Good point. It is true that one has to clarify this important possible influence on the results. We will include a short discussion of this potentially significant source of uncertainty in a revised version of the manuscript.

Author's change: P 14, L 13

Comment 3: Page 12, line 11: why is the 3.5 moment used and not (the most commonly used) 3.67? Could you add a reference?

Reply 3: Thanks for indicating this point. Actually we calculated the rain rate by using the $3.67^{th}$ moment. In a first version however we used the 3.5th moment of the drop size distribution for the rain rate by using the approximation for the fall velocity $v(D) \approx c \cdot D^{0.5}$, which we took from the script of a radar meteorology class. We missed correcting the text accordingly, but will do so in a revised version.

Author's change: P 14, L 10

Comment 4: Page 17, Line 11: A 15.3% reduction in uncertainty when RR0 is conditioned by dBZ1500Rad seems quite low, doesn't it? This proves that radar reflectivity is one of the most important sources of information but that there is a high variability in the Z-R relationship that limits entropy reduction. This explanation is suggested later (experiment 3) but could perhaps already be mentioned here. I think we are also suffering here from the effects of PVR and perhaps even attenuation for high intensities.

Reply 4: We agree. It makes sense to mention the explanation - suggested later in experiment 3 - already in this section. Also, as suggested, we will in a revised version of the manuscript point out that while the radar reflectivity factor is one

of the most important sources of information, there remains a high variability in the Z-R relationship that limits entropy reduction.

Author's change: P 20, L 4

Comment 5: Page 24, Line 3-5: I understand that DKL=3.43 bits and 5.04 correspond to the application of the Z-R Marshall Palmer relationship and that DKL=2.69 and 4.30 correspond to the optimized Z-R relationship. But I don't understand what differentiates 3.43 from 2.69 (and 5.04 from 4.30): what are the differences in terms of condition applied?

Reply 5: It is correct that DKL = 3.43 and Cross Entropy (total uncertainty) = 5.04 bit result from the application of the (deterministic) Marshall-Palmer relationship (red line in Fig. 6), and DKL = 2.69 and Cross Entropy = 4.30 bit result from applying a deterministic Z-R relationship fitted to the available data (black line in Fig. 6). So the difference between 3.43 and 2.69 (and 5.04 and 4.30) comes from using a general Z-R relation (derived from the Z-R data pairs Marshall and Palmer had available) in the first and a fitted-to-the-available data Z-R relation in the second case. So while the Marshall-Palmer relation may be better overall (if used over a large variety of data), the custom-made Z-R relation quite expectedly works better when applied directly to the data it was built from. To us, it was rather surprising how good the Marshall-Palmer relation performed compared to the custom-made relation: It really seems to be a good default.

Comment 6: Page 26, Line 22: after "attributed to the ambiguous relationship between radar reflectivity and rain rate", I think that "due to the natural variability of Drop Size Distribution" may be added.

Reply 6: This is definitely correct. We will add this statement.

Author's change: P 28, L 23

Comment 7: Section 3.6: Why kriging (often used method) was not considered for the raingauge approach?

Reply 7: We agree that it would be interesting to compare our results to established interpolation methods such as kriging. However, performance comparison of interpolation methods was not our main focus in this section. Rather, the goal was to show that our databased approach easily allows joint use of several information sources without many additional assumptions, and secondly to demonstrate that information content is a function of distance between stations. In order to keep the study short, we suggest keeping this section as it is. As a side note: We are currently exploring the potential of data- and information based spatial interpolation in depth, comparing it to a range of established interpolation methods. We hope to submit a related manuscript by mid-2019.

Comment 8: The publication of Cecinate et al (2017) cited on page 2 does not appear in the list of references.

Reply 8: This is right. Thank you for detecting this missing reference. We will add it in a revised version of the manuscript.

Author's change: P 2, L 11 (it was a wrong spelling of the name)

Comment 9: Section 1.1, page 3, is a little "orphaned". It may not be necessary to distinguish this paragraph from the introduction. Its content could appear just before the announcement of the plan (between lines 10 and 11 on page 3).

Reply 9: Another Referee also commented on this section, to which we already replied (Reply 6 to referee #1). In a revised version of the manuscript we will give the block of text before 1.1 a number and subtitle title as well, such maintain the same level of text hierarchy throughout.

Author's change: P 2 , L 6

Comment 10: Table 1: I propose to replace "[min, max]" by"[center of the min bin, center ofmax bin]" or something like that.

Reply 10: We agree, that this would be a more robust value range and not so vulnerable to error prone outliers. On the other hand, we already cut down the outliers and fulfilled a thorough data quality analysis, so that we think, that the total value range is already quite robust and directly shows the total range of possible values.

Comment 11: Figures 4, 5, 6 and 7: I think that the texts of the legends should be enlarged.

Reply 11: We agree that the legends are relatively small. We will ask the HESS typesetting team whether they recommend larger legends for the final layout of the paper and will do so if recommended.

Comment 12: Figure 6: Wouldn't the figure in a log-log frame be more readable and give less weight to very weak reflectivities?

Reply 12: We agree that a log-log display would put more emphasis on the higher reflectivities and rain rates, which are commonly of higher interest than the low ones. However, we would like to keep the figure in its current form as the non-transformed data better reveal the nonlinear power-law shape of the Z-R relation as mentioned in the figure caption and in the text (page 23 lines 10 pp).

Comment 13: Table 3: I propose to replace "RR0" by "RR0Dis" to highlight that disdrometer is used (as for dBZ0Dis).

Reply 13: RR0 does not just contain rain rates measured by the disdrometers, but also by standard rain gauges at Useldange, Roodt and Reichlange (please see the paragraphs 'Distrometer data' and 'Rain gauge data' in section 2.3, and column 'description' in table 1). For this reason, we think the general term 'RR0' is more appropriate than 'RR0Dis'.

Yours sincerely,

Malte Neuper and Uwe Ehret

[revised manuscript text omitted]